# Dysfunction of the episodic memory network in the Alzheimer's disease cascade

René Lattmann [1,2] ✉, Niklas Vockert [1], Jose Bernal [1,2,3,4], Judith Wesenberg [2], Yanin Suksangkharn [1,2], Renat Yakupov [1,2], Hartmut Schütze [1,2], Wenzel Glanz [1], Enise Incesoy [1,2,5], Michaela Butryn [1], Falk Lüsebrink [1], Matthias Schmid [6,7], Melina Stark [6,8], Luca Kleineidam [6,8], Annika Spottke [6,9], Marie Coenjaerts [6], Frederic Brosseron [6], Klaus Fliessbach [6,8], Anja Schneider [6,8], Peter Dechent [10], Klaus Scheffler [11], Stefan Hetzer [12], Alfredo Ramirez [6,8,13,14,15], Christoph Laske [16,17], Sebastian Sodenkamp [16,18], Slawek Altenstein [19,20], Luisa-Sophie Schneider [21], Daria Gref [21], Eike Jakob Spruth [19,20], Andrea Lohse [20], Björn H. Schott [22,23,24], Jens Wiltfang [22,23,25], Ingo Kilimann [26,27], Doreen Goerss [26,27], Ayda Rostamzadeh [28], Josef Priller [19,29,30], Oliver Peters [19,20], Julian Hellmann-Regen [19,31,32], Stefan Teipel [26,27], Michael Wagner [6,8], Frank Jessen [6,13,28], Anne Maass [1,33], Gabriel Ziegler [1,2,35] & Emrah Düzel [1,2,34,35]

Alzheimer's disease (AD) is a major cause of dementia and cognitive decline. Here, we assessed how episodic memory (EM) network dysfunction, a hallmark of AD, is related to the longitudinal progression of AD biomarkers, neurodegeneration and cognition using data from the DZNE DELCODE study. This data set includes over 1000 longitudinal functional magnetic resonance imaging measurements of EM network function. We related activation and deactivation of EM to individual disease progression scores from a disease progression model. Voxel-wise analyses revealed widespread loss of deactivation and activation with disease progression. Trajectories for the loss of deactivation were nonlinear, associated with amyloid- and tau-positivity and visually preceded trajectories of cognitive decline. The relationship between deactivation and cognitive decline was partly independent of neurodegeneration. Our results provide evidence that synaptic dysfunction and neurodegeneration are independent drivers of cognitive decline, providing a rationale for targeting synaptic dysfunction along the AD cascade.

Alzheimer's disease (AD) is one of the leading causes of dementia and cognitive decline[1]. Cognitive decline, and particularly memory loss, is the predominant clinical symptom of AD and the primary endpoint of disease-modifying, symptomatic and risk-reducing treatments[2]. A major challenge is to uncover how cognitive decline relates to the progression of AD. According to the amyloid-cascade hypothesis[3], cognitive decline is the end-stage of a progression from Aβ aggregation to accelerated spreading of hyperphosphorylated tau and consequently to neurodegeneration. However, animal models indicate that cognitive dysfunction can precede neurodegeneration as a result of pre- and postsynaptic dysfunction due to amyloid and tau pathology[4–7]. Here, we addressed this discrepancy leveraging a longitudinal disease progression model (DPM) of human AD in conjunction with measures of cognitive brain activity as a proxy for synaptic dysfunction[8]. This allowed us to model the relationship between cognitive decline and brain activity abnormalities in relation to pathological hallmarks of AD. We utilized cross-sectional and longitudinal measures of CSF-Aβ42/40 ratios, CSF-phosphotau181

(pTau[181]), and neurodegeneration (i.e., volume loss) along the amyloid cascade.

According to an influential model[9], the change towards abnormality in AD biomarkers follows a sigmoidal, monotonically increasing disease progression trajectory, where cognitive decline follows neurodegeneration. The unfolding of this hypothetical cascade over up to two decades[10] has been captured within the amyloid-tau-neurodegeneration (ATN) framework[11]. Investigation of such models requires the conjoint availability of longitudinal CSF biomarkers such as amyloid 42/40 or pTau, volumetry and cognitive markers across the whole AD spectrum, ranging from cognitively unimpaired to mild cognitive impairment and mild dementia[12]. This is highlighted by recent advances from studies using continuous DPMs. Results showed that time frames spanning the whole disease cascade could be estimated from shorter-scale longitudinal data[13–16].

Episodic memory (EM), the ability to recall recent personal experiences[17,18], is critically dependent on the medial temporal lobe (MTL) memory system and structures of the so-called Default Model Network (DMN)[19]. EM is one of the first cognitive faculties to be impaired along the AD cascade[2]. Consequently, a major effort of therapeutic and interventional studies is to slow EM decline or even improve EM function in AD through interventions, including lifestyle[20], pharmacology[21] or transcranial brain stimulation[22]. Improving synaptic function through such interventions could potentially also ameliorate EM impairment[7]. Indeed, targeting the synapse as a potential treatment avenue might prove to be beneficial for neural circuitry and, ultimately, large-scale neural networks, potentially delaying cognitive decline[23]. Thus, uncovering the dynamics of EM network dysfunction across AD progression can inform about the potential to improve EM in patients with AD.

Brain activity related to EM can be measured using task-based functional MRI (fMRI). Task fMRI has been extensively used to investigate EM both in healthy individuals and individuals affected by AD[24]. We used task-fMRI data from a modified version of an incidental learning task[25], in which participants are instructed to classify scenes as indoor or outdoor scenes via a button press. Participants were pre-familiarized with one indoor and outdoor scene. During the task, participants were presented with those familiarized images as well as novel images. Following the fMRI task, participants completed a recognition task during which they had to rate their confidence about their classification of images as novel or familiar. Resulting from the novel and the confidence ratings towards the ratings is a fMRI marker for successful memory encoding which we used in all further analyzes. More details can be found in the "Methods".

Hyper- and hypoactivation-like patterns have been observed in human task-fMRI studies of AD. These studies have reported lower deactivations in regions normally deactivated (also denoted as "hyperactivation" and "reduced deactivation", respectively) during memory tasks like novelty detection or successful encoding, most notably structures of the DMN like the posterior cingulate cortex and precuneus[24,26,27]. For areas activated rather than deactivated during memory tasks (e.g., the inferior temporal lobe and hippocampus), a number of studies have reported hypoactivation[8,28,29]. Explanations for this aberrant activity stem from human and animal research. For example, studies using AD mouse models have shown hyperactivation of neurons in the direct vicinity of soluble and insoluble Aβ deposits compared to neurons distant from deposits in the hippocampus[30,31]. In the face of combined Aβ- and tau pathology, on the other hand, neurons show hypoactivation[5]. One mechanism underlying this activation differences may be synaptic dysfunction. In animal research, synaptic dysfunction has been conceptualized as network dysfunction independent from synapse loss[5]. In line with this rationale, a recent fMRI study has found brain activity abnormality that cannot be explained by neurodegeneration[8].

However, while synaptic dysfunction may be one potential candidate as an explanatory factor of brain activity abnormalities in the human fMRI, further mechanisms have been proposed. For example, studies have provided evidence that vascular pathology[32] or functional connectivity[33] can account for blood-oxygen-level-dependent (BOLD) signal changes. Furthermore, in the cognitive neuroscience literature, dedifferentiation[34] and scaffolding[35] mechanisms are discussed. There, it is still not clear in how far activation abnormalities present as compensatory[35,36] or detrimental. Here, we use atrophy, vascular pathology and functional connectivity in conjunction with activity from EM to investigate EM network abnormalities in human task-based fMRI in AD. Our study builds on reported brain activation abnormalities to model their relationship to AD progression with a DPM.

We generated a computational disease progression marker using a multivariate DPM approach and investigated its relationship with EM-related task-fMRI activation using the largest to date available longitudinal task-fMRI data set in AD. We studied longitudinal changes in task-fMRI activation patterns over multiple follow-ups. Our analyzes were performed in the DELCODE study that covers the full pre-clinical and clinical AD spectrum, including cognitively normal older adults (CN), individuals with subjective cognitive decline (SCD) and patients with MCI and mild dementia of the Alzheimer's type (DAT)[37]. We hypothesized that there would be region-specific associations between encoding-related brain activity and latent disease progression stage. Additionally, following the definition from ref. 8, we hypothesized that AD-related brain activity patterns and brain activity changes would in part indicate synaptic dysfunction by being independent of neurodegeneration.

## Results
### Disease stage associates with several AD-related factors
Baseline demographics are displayed in Supplementary Table 1. Participant selection for the DPM fitting sample was motivated by the ATN framework. In order to obtain a continuous AD-related disease stage score, we used 739 longitudinal ATN biomarker and cognitive measurements from 208 participants with an AD biomarker profile[11] to train the probabilistic DP model (see "Methods" for details; Supplementary Table 2 for baseline characteristics and longitudinal follow-up availability of biomarkers of this subsample). Of note, follow-up years for CSF biomarkers, volumetrics, and cognition were longest in CN, followed by SCD, MCI, and DAT, respectively. For statistics of those differences, please see Supplementary Table 2. The model results were two-fold. First, a posterior probability distribution for the empirical biomarker disease progression curves was obtained. For visualization, we plotted the average GPs after randomly sampling from this posterior probability distribution 200 times (Fig. 1a, and Supplementary Fig. 1). Second, each participant was assigned a continuous disease stage value on an arbitrary scale in years. The obtained disease stages are probabilistic estimates given the available biomarker and cognitive test data of the patient. Thus, the disease stage values characterize the relative rather than absolute stages, reflecting individual differences in the analyzed sample on the latent disease time axis span by the DPM. To aid clinical interpretation, we anchored the disease stages around the median disease stage of healthy controls with amyloid positivity (CN A+T-). Concretely, a participant's disease stage is given as the minimum in the negative log-likelihood function over the disease stages given all available participant data. In this regard, more positive disease stages are related to objective memory impairment, amyloid positivity, and conversion from CN/SCD to MCI or AD (Supplementary Fig. 2). Bivariate correlations between DPM markers and obtained disease stages are provided in Supplementary Fig. 3. Inspecting timepoints of fastest changes, we noted that CSF biomarkers became abnormal

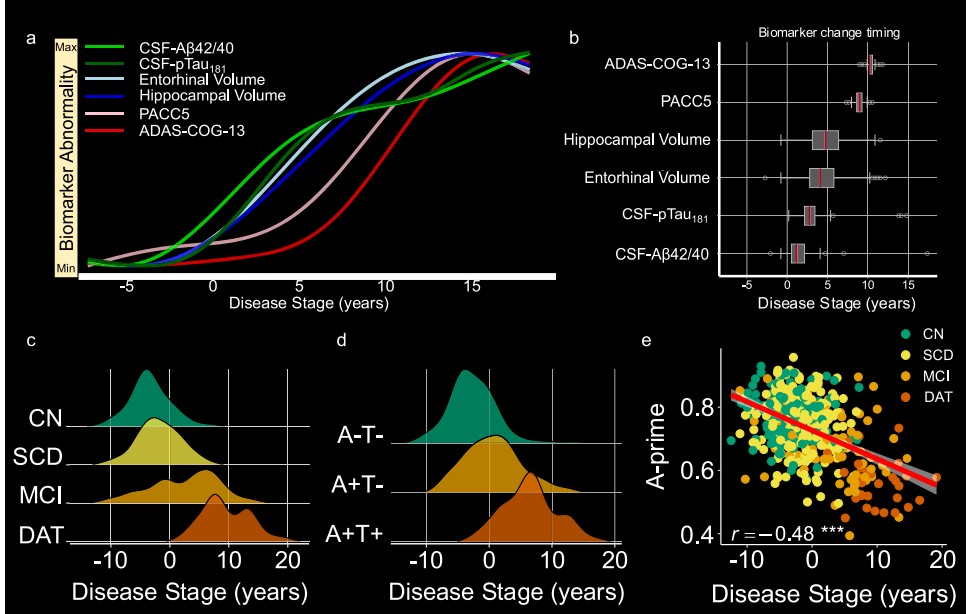

**Fig. 1 | Disease progression curves and association with AD-related variables.**
**a** GP-based DPM used in this study comprising longitudinal CSF (Aβ42/40, pTau$_{181}$), volumetric MRI (hippocampal, entorhinal volume), and PACC5 as well as ADAS-COG-13 cognitive score within the DELCODE cohort using 787 available data points from 208 participants (80 CN, 57A+ SCD, 44A+ MCI, 27A+ DAT). Different colour groups pertain to the different biomarker groups. **b** Timepoints of fastest changes derived from temporal derivatives of the empirical biomarker progression curves from the model posterior distribution. The timepoint of fastest change was sampled $n = 200$ times from the model posterior distribution by obtaining 200 independent biomarker progression curves and calculating their temporal derivatives. Box plots denote the interquartile range (IQR; third quartile (Q1) – first quartile (Q1)) around the median. Whiskers show minima and maxima as defined by X >= Q1 − 1.5 * IQR or X <= Q3 − 1.5 * IQR, respectively. **c** Ridgeline plots of cross-sectional disease stages in diagnostic groups (colours) of the full analysis sample ($n = 493$). **d** Ridgeline plots of cross-sectional disease stages in AT classification subgroups (colours) of AD pathology ($n = 222$). **e** Association between the estimated disease stage and memory performance during the task-fMRI session residualized for age, sex, and education ($n = 493$; predicted values are displayed as mean ± SEM). ***$p$(two-sided) = $1.522 \times 10^{-29}$. ADAS-COG-13: Sum score of the cognitive subscale of the Alzheimer's disease assessment scale version 13, CN cognitively healthy older adults, SCD subjective cognitive decline, MCI mild cognitive impairment, DAT dementia of the Alzheimer's type, PACC5 Preclinical Alzheimer's cognitive composite. Source data are provided as a source data file.

first, followed by brain volume estimates and, ultimately, cognitive performance (Fig. 1b). Investigating the associations of our disease stage scores with demographics in a multiple regression analysis, we found demographics to contribute significantly to the disease stage variability ($F(3,489) = 42.56$, $p = 1.884 \times 10^{-24}$, adjusted $R^2 = 0.202$). Specifically, higher age was indicative of a more advanced disease stage ($t(489) = 9.148$, $p = 7.836 \times 10^{-19}$, $\eta^2_{partial} = 0.17$, 95% CI = [0.13, 1]), whereas higher education was related to earlier disease stages ($t(489) = -4.979$, $p = 4.438 \times 10^{-7}$, $\eta^2_{partial} = 0.05$, 95% CI = [0.02, 1]). Sex showed no significant association with estimated disease stage ($t(489) = 0.637$, $p = 0.524$, $\eta^2_{partial} = 0.0005$, 95% CI = [0, 1]). All further analyzes with disease stage consider the age-, sex-, and education-corrected disease stage values.

Next, associations with diagnostic groups and AT biomarker categories were investigated. Diagnostic group membership was significantly related to disease stage ($F(3,479) = 133.11$, $p = 1.016 \times 10^{-62}$, $\eta^2_{partial} = 0.46$, 95% CI = [0.4, 1]), reflecting the a priori assignment of participants to diagnostic groups at baseline. *Post-hoc* analysis revealed the following ordering of disease stage for baseline diagnosis: CN < SCD < MCI < DAT (Fig. 1c, and Supplementary Table 3). Additionally, when analyzing disease stage differences over the AT criteria, biomarker groups were significantly associated with disease stage ($F(2,209) = 92.11$, $p = 2.070 \times 10^{-29}$, $\eta^2_{partial} = .47$, 95% CI = [0.39; 1]). Ordering of disease stages by AT groups was as follows: A-T- <A + T- <A + T+ (Fig. 1d, and Supplementary Table 4).

Lastly, we were interested in the association between disease stages and a fMRI task performance marker (A′ – "A prime") not used in model fitting. Task performance was measured as the area under the curve of hits versus false alarms in the recognition task 70 min post fMRI task (see ref. 38 for details). While a score of 0.5 would indicate guessing randomly, larger scores reflect better performance. More details can be found in the "Method" section. As expected, participants further in disease progression showed worse A′ scores of fMRI task performance ($r(491) = -0.48$, 95% CI = [−0.54; −0.40], $p = 1.522 \times 10^{-29}$; Fig. 1e) accounting for age, sex, and education.

## Encoding-related brain activity changes with progression towards AD

After quantitatively establishing a marker of a continuous disease score characterizing progression towards AD in the previous section, we investigated encoding-related activity differences as a function of disease progression. Such differences during the course of AD are potentially reflective of disease-related functional reorganization and adaptation. In the whole sample ($n = 493$), successful memory encoding was related to activation of a large network including lingual gyri, occipital, and prefrontal regions, while widespread deactivations were observed in the precuneus, posterior cingulate cortex, inferior parietal lobule, and fronto-temporal regions (Fig. 2a), replicating previous observations in the same dataset[26,27,39] and in independent cohorts[40-43]. We observed that more advanced disease scores were related to hyperactivation in the precuneus, inferior parietal lobule, and posterior cingulate cortices bilaterally, as well as the anterior cingulate cortex and superior frontal gyrus (Fig. 2b, orange colors, Family-Wise error (FWE) corrected with $p_{FWE} < 0.05$), reflecting similar previous observations based on the novelty contrast of the same paradigm[28]. Additionally, we found right-lateralized reductions in encoding-related

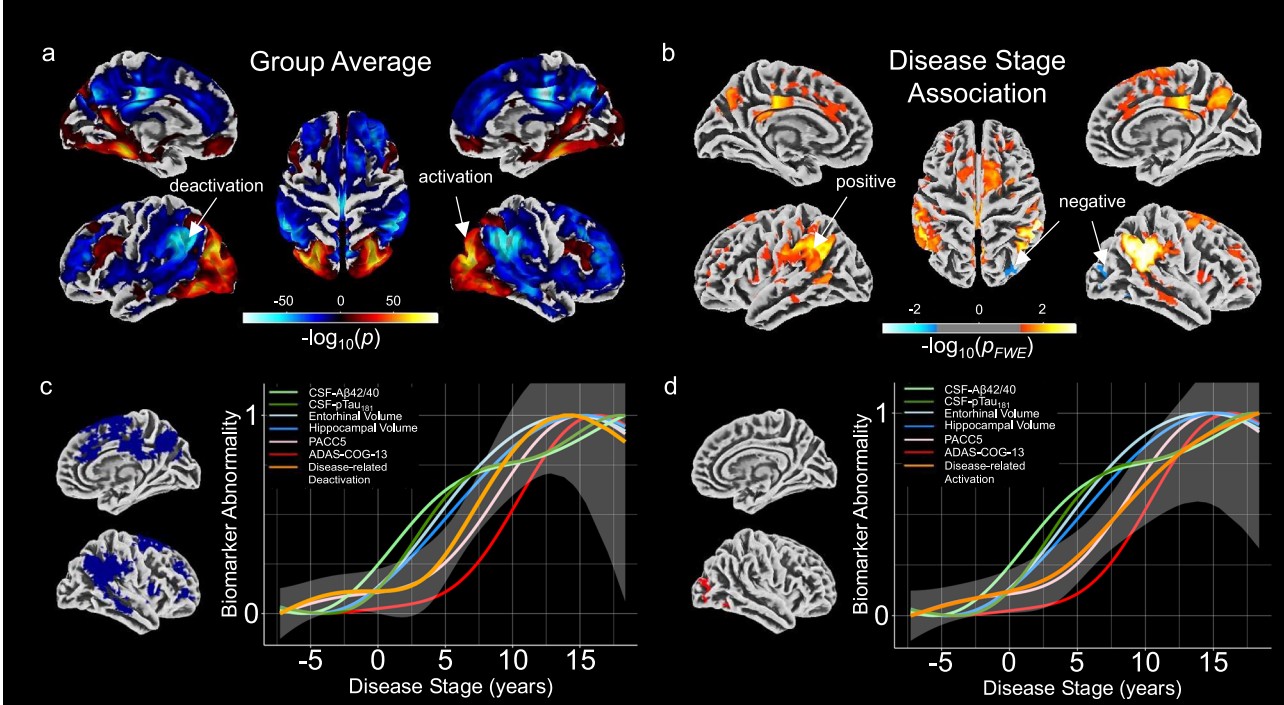

**Fig. 2 | Association between encoding-related activity and disease stages.**
**a** Surface representation of the average activation and deactivation for successful memory encoding extracted from the second-level sandwich estimator (SwE) model ($n = 493$). Values were thresholded at $p < 0.00001$. In this figure, we combined results from the two separate one-sided $t$-tests testing for activation and deactivation effects separately. While red colours pertain to regions of significant activation, blue colours pertain to regions to significant deactivation, respectively. **b** Linear associations of disease stage with successful memory encoding related activation and deactivation from the second-level SwE model. Results from two-separate one-sided $t$-tests testing for both directions were obtained using the Wild bootstrap method with 1000 repetitions adjusting for multiple comparisons using family-wise error at $p_{FWE} < 0.05$. Here, warmer colours represent significant regions

of a linear positive relationship between disease stage and deactivation, while colder colours represent linear negative relationships between disease stage and activation regions. Note that we did not find negative linear relationships in deactivation regions and vice versa. **c** Relative position of biomarker abnormality differences for age-corrected encoding-related deactivation in relation to the biomarkers from our data-driven DPM approach ($n = 493$). FMRI abnormality starts in the earliest disease stages. Towards disease stage 0, a small plateau can be observed before increasing in abnormality together with volumetric biomarkers. **d** Relative position of biomarker abnormality change for the age-corrected successful memory encoding related activation. For activation, we only observed a linear increase in abnormality over the disease progression. Source data are provided as a source data file. **c, d** Colours represent the different biomarkers.

activation with advancing disease progression within the middle occipital gyrus and inferior temporal gyrus (Fig. 2b, blue colors, $p_{FWE} < 0.05$). Although we tested for non-linearities, we did not find indications for inverted U-shaped associations between disease stage and successful memory encoding. Additional u-shaped associations are reported in Supplementary Fig. 4. In summary, we found hyper-activations (or reduced deactivations) as well as a reduction in activation during successful memory encoding with further disease progression.

**Towards task-activity based AD biomarker progression curves**
Since the progression of activity changes in patients transitioning towards AD exhibited unexpected dynamics (e.g., in early stages), we next assessed activity progression curves with a more flexible curve fitting approach using $n = 493$ data points (i.e., one value per participant). Obtained progression trajectories from age-corrected mean activity in clusters from the above analyzes are presented in Fig. 2c, d. Deactivation followed a non-linear progression trajectory, which was characterized by an initial increase of abnormality in the earliest disease stages, followed by the most pronounced changes in later disease stages ($F_{non-linear}(1.739, 489.216) = 4.567$, $p_{raw} = 0.01149$, $p_{Holm} = 0.02298$, $\eta^2_{partial} = 0.016$, 95% CI = [0; 1]; Fig. 2c). These later changes overlapped visually with changes in hippocampal and entorhinal cortex volume. Cognition started to show abnormalities after memory-related fMRI deactivations (time points of fastest change

(in years) - Aβ42/40: 1.17; pTau$_{181}$: 2.79; Entorhinal Volume: 4.09; Hippocampal Volume: 5.38; EM Deactivation: 6.36; PACC5: 8.95; ADAS-COG-13: 10.57; Fig. 1b). Disease-related changes in fMRI activations resembled a monotonic increase as well, yet the non-linear effects were not significant ($F_{non-linear}(1.368, 489.632) = 2.931$, $p_{raw} = .07389$, $p_{Holm} = 0.07389$, $\eta^2_{partial} = 0.00812$, 95% CI = [0; 1]; Fig. 2d). In Supplementary Fig. 5, we provide a comparison between raw and age-corrected activation values and how this influences the trajectory estimation.

**Subsequent memory is associated with individual biomarkers**
The disease stage is a marginal score combining the influence of several components into one single index. Thus, we were further interested in individual contributions of biomarkers towards activity variability beyond demographic information (age, sex, and years of education). First, AT group membership was significantly associated with memory-related deactivation ($F(2,157) = 13.35$, $p = 4.421 \times 10^{-6}$, $\eta^2_{partial} = 0.15$, 95% CI = [0.07; 1]). *Post-hoc* tests revealed that combined amyloid and tau positivity (A+T+) was related to reduced deactivations compared to the other groups (A-T-, $t(157) = 4.981$, $p = 8.260 \times 10^{-7}$, and A+T-, $t(157) = 4.397$, $p = 1.007 \times 10^{-5}$). A correction for hippocampal and entorhinal volume ($F(2,155) = 4.82$, $p = 0.009$, $\eta^2_{partial} = 0.06$, 95% CI = [0.01; 1]) did not change the association (A-T-, $t(155) = 2.81$, $p_{Holm} = 0.007$, and A+T-, $t(155) = 2.46$, $p_{Holm} = .022$; Fig. 3a, blue colors). This was also true for a direct test via mediation analyzes. An additional

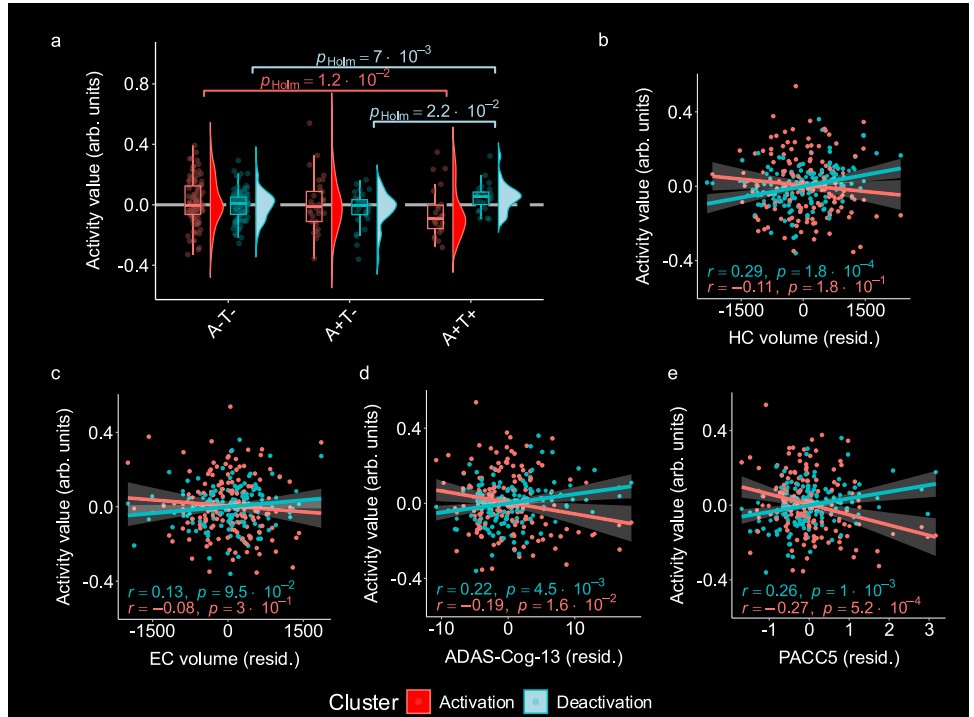

**Fig. 3 | Distinct contributions of DPM biomarkers to disease-related activation variability in successful memory encoding. a–e** We used a subsample of $n = 160$ individuals ($n_{A-T-} = 104$, $n_{A+T-} = 32$, $n_{A+T+} = 24$) with available fMRI and complete biomarker data (i.e., CSF, volume, and cognition) to determine unique contributions of DPM biomarker groups to disease-related activation variability. Warmer colours represent mean values from activation regions while colder colours represent mean values from deactivation regions. **a** Association between AD biomarker staging groups and encoding-related activity, corrected for differences in volume using ANCOVA models with activity values as dependent variables and biomarker group as independent variable, controlling for hippocampal and entorhinal volume. Post-hoc $t$-tests were adjusted for multiple comparisons using Bonferroni–Holm. Box plots denote the interquartile range (IQR; third quartile (Q1)

– first quartile (Q1)) around the median. Whiskers show minima and maxima as defined by X > = Q1 - 1.5 * IQR or X < = Q3 - 1.5 * IQR, respectively. **b–e** Semi-partial correlations between activity and biomarkers of atrophy and cognition. Hippocampal and entorhinal volume are corrected for CSF Aβ42/40 ratio and pTau$_{181}$. Cognition variables are corrected for hippocampal and entorhinal volume. All variables are additionally corrected for age, sex, and education. Data are presented as mean ± SEM. Significance was tested using two-sided $t$-tests. Raw $p$-values are reported. arb. units arbitrary units, ADAS-COG-13 = Sum score of the cognitive subscale of the Alzheimer's disease assessment scale version 13, EC entorhinal cortex, HC Hippocampus. PACC5 Preclinical Alzheimer's cognitive composite. Source data are provided as a source data file.

correction for both volumes and both cognitive measures from our DPM ($F(2,153) = 3.12$, $p = 0.047$, $\eta^2_{partial} = 0.04$, 95% CI = [0.0003; 1]) did not change the association. For memory-related fMRI activation (task-positive regions), we also found a significant association with AT group membership ($F(2,157) = 6.212$, $p = 0.003$, $\eta^2 = 0.07$, 95% CI = [0.02; 1]). Post-hoc tests revealed that this was driven by the lower activation in the A + T+ compared to the A-T- group ($t(157) = -3.465$, $p = 0.002$). While additional control for hippocampal and entorhinal volume did not change the association, ($F(2,155) = 4.31$, $p = 0.015$, $\eta^2_{partial} = 0.05$, 95% CI = [0.01; 1]; Post-hoc analysis (A-T-), $t(155) = -2.14$, $p_{Holm} = 0.012$, Fig. 3a, red colors), AT group membership was not related to variability in successful memory-related activation when controlling for volume and cognition measures from our DPM combined ($F(2,153) = 1.56$, $p = 0.213$, $\eta^2_{partial} = 0.02$, 95% CI = [0; 1]).

Next, we calculated semi-partial correlation coefficients for the association of DPM biomarkers with activation and deactivation values, correcting for the influences of other biomarkers in the DPM, respectively (Fig. 3b–e). For this, we multiplied volumes and the PACC5 score by −1, such that higher values in all DPM variables indicate higher amounts of pathology. We observed that, when controlling for CSF biomarkers, hippocampal volume was positively associated with mean cluster deactivation ($r(158) = 0.29$, $p = 1.80 \times 10^{-4}$, 95% CI = [0.14; 0.43], Fig. 3b) but not with activation ($r(157) = -0.11$, $p = 0.18$, 95% CI = [−0.26; 0.05], Fig. 3b). Entorhinal cortex atrophy was not correlated with fMRI activity (all $p > 0.05$, Fig. 3c). Regarding cognitive performance, we

found that encoding-related deactivation was significantly positively correlated with ADAS-COG-13 ($r(158) = 0.22$, $p = 4.5 \times 10^{-3}$, 95% CI = [0.07; 0.37]; Fig. 3d) and with PACC5 ($r(158) = 0.26$, $p = 1.02 \times 10^{-3}$, 95% CI = [0.11; 0.40]; Fig. 3e) when the influence of hippocampal and entorhinal volumes was controlled for. Encoding-related activation was significantly negatively correlated with ADAS-COG-13 ($r(157) = -0.19$, $p = 0.016$, 95% CI = [−0.37; −0.08]; Fig. 3d) and with PACC5 scores ($r(157) = -0.27$, $p = 5.2 \times 10^{-4}$, 95% CI = [−0.41; −0.12]; Fig. 3e) when controlling for both volumes. In summary, the emergence of tau and hippocampal volume provides unique explanatory variance to disease-related fMRI activation differences. Additionally, beyond the influences of volume, brain activity explained cognitive performance differences.

**Investigating third variables to disease-related brain activation**
In order to discern whether the associations survive the correction for additional third variables, we performed parallel mediation analyzes including volume, white matter hyperintensities (WMH), and effective connectivity. Effective connectivity values were obtained from a recent study utilizing dynamic causal modeling (DCM)[44]. WMH were log-transformed and all variables entering the model were corrected for baseline age, sex, and years of education a priori. Supplementary Fig. 5 provides an illustration of this mediation analysis. The complete model outputs including fit indices can be found in the Supplementary Tables 5 and 6. For disease-related activation, hippocampal volume

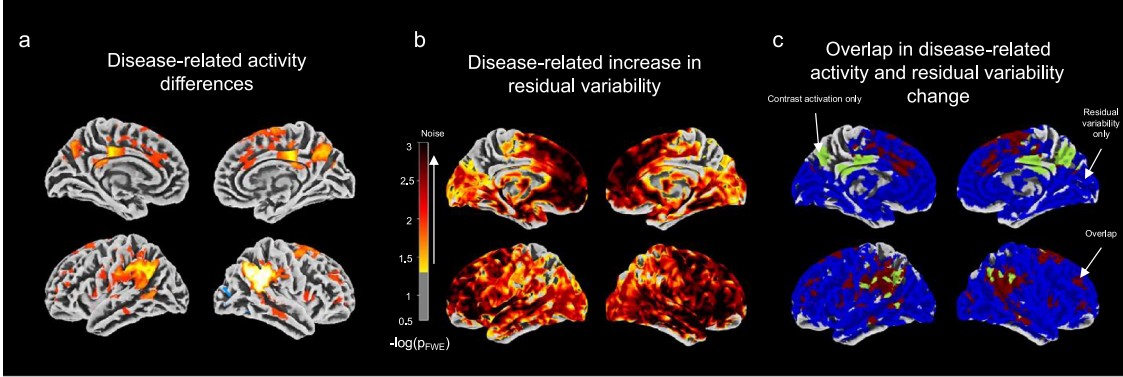

**Fig. 4 | Larger residual variability in the fMRI is related to advanced disease stages and overlaps with disease-related activity differences. a** For completeness, we plot here results from Fig. 2b, highlighting significant regions of activation differences across the disease stages. Multiple comparisons adjusted for by family-wise error at $p_{FWE} < 0.05$. **b** We observed a positive association between residual variability and disease stages. Results were obtained using Wild bootstrap with 1000 repetitions adjusted for multiple comparisons using family-wise error thresholded at $p_{FWE} < 0.05$ in the full sample of $n = 493$ individuals. We report on the one-sided t-contrast results. Colours represent the magnitude of the linear increase in residual variability across disease stages. **c** Regions of positive association between residual variability and disease stage overlaps (red regions) with activity differences. Additionally, we identified regions in which effects were unique highlighted in green (for activity only) or blue (for residual variability only). Source data are provided as a source data file.

(indirect path: $z = 2.102$, $p = 0.036$) and effective connectivity (indirect path: $z = 2.057$, $p = 0.040$) mediated the association between activation and cognition. Entorhinal volume (indirect path: $z = 1.580$, $p = 0.114$) and WMH (indirect path: $z = 1.023$, $p = 0.306$), on the other hand, did not significantly mediate the association. Even if mediators were taken into account, higher levels of cognition were related to higher activation (direct path: $z = 4.117$, $p < 0.001$). For disease-related deactivation (Supplementary Fig. 4B), again, hippocampal volume (indirect path: $z = -2.211$, $p = 0.027$) and effective connectivity (indirect path: $z = -2.354$, $p = 0.019$) partially mediated the association between deactivation and cognition (direct path: $z = -4.799$; $p < 0.001$), but not WMH (indirect path: $z = -1.414$, $p = 0.157$) or entorhinal volume (indirect path: $z = -1.814$, $p = 0.070$).

Finally, we also tested for whether differences in residual variability could explain our fMRI results. Instead of the contrast images as in Fig. 2b, we submitted the residual variability maps from the first-level fMRI modeling to the second-level SwE utilizing the same design matrix as above. We found that there was a wide-spread increase in residual variability over the disease stages after controlling for FWE (threshold $p < 0.05$; Fig. 4b). The disease-related differences also partly overlapped with the disease-related activation results from section "Encoding-related brain activity changes with progression towards AD" (Fig. 4c). To determine whether noise or the activity values predominates in the disease stage association, we extracted baseline contrast values and noise estimates from the overlapping regions and submitted them to a multiple regression analysis controlling for age, sex, and years of education with A' as the dependent variable. Result tables can be found the Supplementary Tables 7 and 8.

We found the association between activity and cognitive performance (activation regions: $t(428) = 7.190$, $p = 1.455 \times 10^{-12}$, $\eta^2_{partial} = 0.14$, 95% CI = [0.10; 1]; deactivation regions: $t(428) = -5.210$, $p = 1.471 \times 10^{-7}$, $\eta^2_{partial} = 0.08$, 95% CI = [0.04;1]) was stronger than the association between residual variability and cognitive performance (activation regions: $t(428) = -2.908$, $p = 0.004$, $\eta^2_{partial} = 0.04$, 95% CI = [0.01; 1]; deactivation regions: $t(428) = -2.666$, $p = 0.008$, $\eta^2_{partial} = 0.04$, 95% CI = [0.02; 1]). In summary, activity is significantly related to cognitive variability even when accounting for structural, vascular, effective connectivity, or residual variability.

### Longitudinal activation change is moderated by disease stage

If longitudinal data were reflective of our empirical disease curves for fMRI from above, this would provide further leverage for establishing fMRI as a biomarker in AD. To this end, we hypothesized that cross-sectional disease stage would moderate longitudinal changes in successful encoding-related activity across follow-ups, particularly in deactivation. For this analysis, we used longitudinal available fMRI data from four time points. Mean follow-up availability was different across the groups with more data available for CN than any other group ($F(3,132.41) = 25.189$, $p = 5.870 \times 10^{-13}$, $\eta^2 = 0.10$, 95% CI = [0.06; 1]; Supplementary Table 1). Random effect variances in the random-intercept-random-slope models were very close to zero, which is why we report on the random intercept models only. A potential reason may be the inherent problem of large intra-individual variability known to occur in longitudinal task-fMRI[45,46]. The corresponding tables for the random-intercept models can be found in the Supplementary Tables 6 and 7. As expected, we found change over follow-ups in encoding-related deactivation to be significantly moderated by disease stage ($t(346.969) = 2.716$, $p_{raw} = 0.00693$, $p_{Holm} = 0.01386$, $\eta^2_{partial} = 0.02$, 95% CI = [0.00323;1]; Fig. 5a). On the other hand, this was not the case for encoding-related activations ($t(317.37) = 1.473$, $p_{raw} = 0.141758$, $p_{Holm} = 0.141758$, $\eta^2_{partial} = 0.006$, 95% CI = [0;1]; Fig. 5b). Finally, we investigated associations between longitudinal changes of both encoding-related fMRI activation and disease progression. To this end, we correlated individual-level activation and deactivation slopes over follow-ups with intra-individual slopes of the disease stage variable. Neither changes in activation nor in deactivation were significantly correlated with the changes in disease stage in our sample (all $p > 0.05$).

### Discussion

In the present study, we could relate activity of EM network to an AD DPM. Our findings show that neural activity in the EM network changes alongside disease progression and seems to follow MTL atrophy. Consistent with this potential cascade, our longitudinal fMRI findings show that loss of deactivation was most prominent at later disease stages. However, our findings also indicate that the relationship between EM network dysfunction and hippocampal volume on the one hand, and tau pathology, on the other hand, is partly independent. This independence indicates that memory function may be associated with multiple processes along the AD cascade, including synaptic dysfunction and neurodegeneration. We additionally showed that those results were only in part mediated by hippocampal volume and effective connectivity. Finally, neural activity in the EM network predicted cognition independent of AD biomarkers and MTL volume.

We started our analysis by first conceptualizing a continuous Bayesian DPM[14] using longitudinal CSF, volume and cognition data based on the ATN framework. We obtained biomarker progression curves and disease stage scores. The former resulted in a disease time frame which reflects the successive increase in biomarker abnormality of ATN and cognition data in the DELCODE sample. The disease stages provide probabilistic information about the position of a participant on this arbitrary disease time frame, given available longitudinal ATN and cognition data. We found that our disease time frame was around 20 years (Fig. 1a, b). This is in line with previous conceptualizations showing that AD may unfold over more than two decades[10]. As predicted, disease stage scores were associated with clinical groups, ATN staging and memory accuracy in the fMRI task (A prime) (Fig. 1c–e). Finally, the inferred time points of fastest change in the data-driven model (Aβ42/40: 1.17; $p$Tau$_{181}$: 2.79; Entorhinal Volume: 4.09; Hippocampal Volume: 5.38; EM Deactivation: 6.36; PACC5: 8.95; ADAS-COG-13: 10.57; Fig. 1b) do align with previously hypothesized trajectory patterns[47] as well as findings from a previous DPM study[48].

After establishing an AD progression marker using the DPM framework, we investigated the relationship between fMRI activations during EM encoding and our DPM scores. Globally, our results indicate a conjoint reduction of encoding-related activations and deactivations as individuals progress along the AD trajectory (Fig. 2, results section "Encoding-related brain activity changes with progression towards AD"). Thus, individuals in our sample show less pronounced brain activity differences between successfully memorized and later forgotten stimuli as cognition declines, an observation in line with the recently reported risk-dependent reduction of fMRI subsequent memory effects across the Alzheimer's risk spectrum[26]. Additionally, our results may be in line with the dedifferentiation hypothesis of cognitive aging[34,49]. According to the dedifferentiation hypothesis of cognitive aging, brain areas might fail to respond in a selective and specific manner towards different stimuli. Since the underlying successful memory contrast in our study is an indication of differential activation towards successfully encoded vs. non-encoded items, it may be plausible that the EM network experiences a reduction in specificity to successfully encoded vs. non-encoded stimuli, giving rise to cognitive decline via dedifferentiation. Similar results were reported earlier in a study investigating functional connectivity in preclinical AD[50]. However, we did not find other regions outside the episodic memory network suddenly showing a subsequent memory effect along the disease cascade. Thus, the evidence for dedifferentiation in our analysis remains incomplete. Further, our results indicate that the effect sizes for deactivation were larger than for activation, resonating with previous studies that show that task-negative areas overlapping with the DMN[19] are affected in AD[28,51,52]. Reduced DMN deactivations during episodic encoding can be found even in healthy older adults[40,41], and they have been associated with relatively poorer memory performance[25,42].

Given the non-linear course of biomarker abnormalities from our DPM, we next explored potential non-linear associations of regional deactivations and activations (from the fMRI analysis) with disease progression. Thereby, we could identify time points of the fastest brain activity change. Our findings suggest that fMRI deactivations may become abnormal before overt cognitive deficits, but after CSF biomarkers and MTL volume (Fig. 2c). Importantly, this was corroborated by our longitudinal results, showing more pronounced effects in change of deactivation over time for later-stage participants than for earlier-stage participants (Fig. 5a). This is in contrast to studies suggesting that EM network dysfunction may appear as one of the earliest symptoms in the cascade[53].

Next, we assessed individual contributions of each DPM marker to activity differences (Fig. 3). Our results suggest that synaptic dysfunction and neurodegeneration may be independent drivers of EM network dysfunction. We found that the emergence of tau was related

to a reduction in deactivation and activation, respectively, even when controlling for hippocampal and entorhinal volume. This is in line with a previous study showing that novelty-related activation was related to tau pathology while correcting for hippocampal volume in humans[8]. Moreover, our results resonate with previous studies reporting that tau oligomers can directly lead to synaptic dysfunction (for a review, see ref. 23), particularly when co-occurring with amyloid pathology[5,8]. These results indicate that deactivation changes in the EM network in AD are partly a consequence of MTL atrophy and partly reflect tau-mediated synaptic dysfunction independent of atrophy. Finally, deactivation changes in EM were related to cognition independently of hippocampal and entorhinal volume, suggesting a relationship between synaptic dysfunction and cognition[7].

However, since synaptic dysfunction may not be the only mechanistic explanation behind our results, we additionally investigated the influence of other potential mechanisms, including functional connectivity, WMH, and neural noise. We found that, even when including functional connectivity and WMH volume into a parallel mediation model in conjunction with measures of MTL volume, the association between encoding-related activation and cognition remained. Therefore, it appears unlikely that our disease-related activation was driven by white matter damage or functional connectivity. With regards to the investigation of noise or unexplained variability in the fMRI, we found that overlapping disease-related differences in (de-)activation and noise both explained cognitive variability, yet the effect sizes for BOLD activity were much larger than for noise when incorporated into the same model. Specifically, we found that unexplained variability increased over AD disease stages. Those findings may be related to studies showing that AD is characterized by a reduction in neural selectivity[54], with potential mechanistic explanations being related to excitation-inhibition imbalance[55], dedifferentiation[34], or a breakdown of scaffolding[35] mechanisms.

Taken together, our findings provide longitudinal evidence about the relationship between EM network dysfunction and the Alzheimer's disease cascade. Our DPM, our analysis of non-linearity, the results from the dependency analysis, and the results from our longitudinal analysis enable us to position EM network dysfunction within the ATN framework. As illustrated in Fig. 6, our data confirm a neuropathological cascade from amyloid pathology (A), to tau pathology (T), to neurodegeneration (N) and finally to cognitive impairment (C). We suggest that EM network dysfunction (E) partly directly follows T and is partly related to N. We also suggest that E regularly precedes C. This extended ATN model has implications for the treatment of AD, because it identifies network dysfunction as a probable cause for cognitive impairment that is independent of atrophy and therefore potentially reversible. Our extended ATN model thus suggests that there is a time window within the AD cascade in which a reduction of amyloid and tau pathology can improve cognition, even if there is already some degree of irreversible neurodegeneration (Fig. 6).

This study is not without limitations. First, a small number of participants contributed data to the DPM beyond 15 years of estimated disease duration. Due to the low number, and in light of the previously reported poor expression of fMRI episodic encoding effects in individuals with manifest AD[26,27], results particularly for the very end of the disease stage spectrum need to be interpreted with caution. Additionally, model fit in DPMs is dependent on the measurement noise of utilized biomarkers. A previous study showed that measurement noise in CSF biomarkers can obscure group-level trends such that biomarker trajectories could not be fitted[12]. Furthermore, large cross-sectional variability can obscure longitudinal trends which may result in a greater the emphasis of cross-sectional in comparison to longitudinal information in the fitting process of the DPM. Second, we did not correct volumes for age in our DPM approach before entering the DPM algorithm. This was done to preserve the disease-related variability in

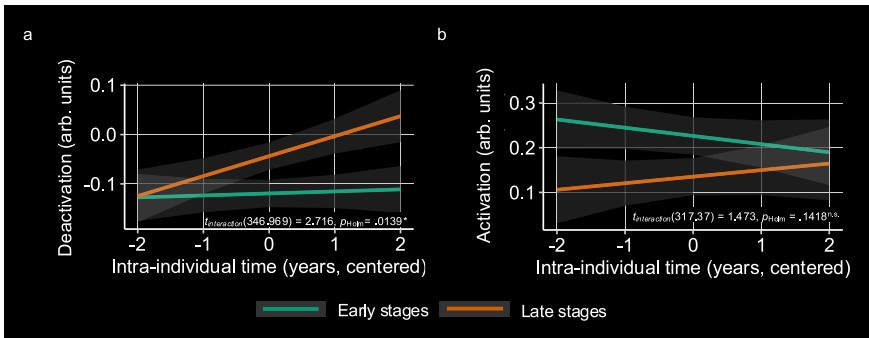

**Fig. 5 | Association between disease stage and longitudinal change of successful memory encoding–related activation and deactivation.** For this analysis, longitudinal fMRI data from $n = 166$ individuals with available AD biomarker information was used ($n_{A-T-} = 108$, $n_{A+T-} = 33$, $n_{A+T+} = 27$). For visualization of the interaction, participants were grouped into early and late stages based on the median disease stage (−3.37) in the sample, such that each participant with a disease stage smaller than −3.37 was defined "in early stages" whereas every participant with a higher disease stage value was defined as "in late stages" of AD. Colours pertain to the different groups. **a** Interaction effect Time × Disease Stage on disease-related encoding deactivation. **b** The same interaction effect as tested in **a**, but for disease-related encoding activation. Here, change over follow-ups is not significantly moderated by disease stage. Statistical significance of parameter estimates were tested using two-way one-sample $t$-tests. $P$-values were corrected using the Bonferroni–Holm procedure. Predicted values are displayed as mean ± standard error. * $p_{Holm} < 0.05$; n.s. $p_{Holm} > 0.05$. Source data are provided as a source data file.

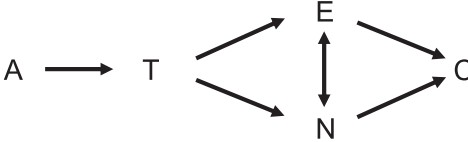

**Fig. 6 | An extended ATN model including episodic memory network dysfunction.** Our results confirm previous hypothetical conceptualizations of an AD cascade with cognitive impairment (C) being preceded by neurodegeneration (N), tau pathology (T) and amyloid pathology (A). Here, we additionally incorporate episodic memory dysfunction (E) as being partly driven by T and partly by N. E may furthermore result in more N. More explanations in the text.

our sample, since groups of MCI and AD were older than the healthy subsample. Third, we used CSF biomarkers instead of regional biomarker load (PET) resulting in a global estimate for amyloid and tau biomarker status. In particular, a previous study showed that local tau depositions are associated with local changes of activation in EM systems[56]. Further, essays of $p$Tau$_{217}$ were not available in the data set. It has been recently shown that $p$Tau$_{217}$ outperforms $p$Tau$_{181}$ in terms of diagnostic performance[57,58]. Of particular note, we did not observe sex differences in our study in terms of disease stages, even though a large body of research suggests differential cognitive decline[59,60], biomarker accumulation[61,62], and incidence rates for females compared to males[63]. This may be related to a sampling effect as both, our fitting sample (Supplementary Table 1) and overall analysis sample (Supplementary Table 2) only presented with minor differences in sex distributions across diagnostic groups. Further, based on the DPM conceptualization, all participants are hypothesized to progress through the disease with the same speed[14]. Thus, the disease stages are informative about the relative position on the latent disease time axis but do not inform about differential speed of decline. Fourth, first-level fMRI contrast images were not corrected for cerebrovascular contributions in our study[64]. For instance, it was recently shown that variability in cerebrovascular reactivity has an influence on the BOLD response[65] and thus, on the estimation of effects and, ultimately, the activation values from our first-level models. Fifth, as the activation differences along the disease progression vanished when excluding AD participants from the model and since this subsample is prone to significantly encoding fewer items, we cannot rule out that the reduction in activation may stem from a regression towards the mean. On the other hand, a loss of power due to an exclusion of participants

may also explain the null findings for activation when excluding AD participants. Additionally, we aimed at mitigating this already by excluding participants with a response bias prior to entering them in the analysis[39]. Seventh, aging represents the primary risk factor for AD, and advancing age is tightly coupled with both increased incidence and greater disease burden[66]. Accordingly, a complete disentanglement of effects attributable to AD progression from those attributable to normative aging is not only challenging but may obscure meaningful sources of variance within patient samples. In the present study, we sought to mitigate this confounding by statistically residualizing for chronological age across all analyzes. However, this approach assumes a linear relationship between age and neural measures, an assumption that may not fully capture the complex, potentially nonlinear interactions between aging and disease-related processes. A more precise characterization of these interactions was beyond the scope of the current manuscript but represents an important direction for future work. Finally, the fMRI-related activation was characterized by large intra- and interindividual variability, such that our results need to be interpreted with caution. More specifically, thus, in order to confirm our extended ATN model, larger samples with longitudinal task-fMRI, CSF measurements, regional biomarker load (PET), and cerebrovascular data are needed. Future research might therefore examine individual change-change associations between activation and local deposition of biomarkers using PET. Additionally, the DPM approach we focused on deviates from parametric DPMs that impose different and often stronger assumptions on trajectories (e.g., sigmoidal, piecewise linear or dynamical approaches; e.g.,[12,15,67]). The Gaussian process framework of ref. 14 provides a non-parametric, continuous representation of disease evolution, allowing relatively flexible capture of non-linear abnormality curves without predefining functional forms. This distinguishes it also from popular event-based subtyping models such as SuStaIn[68,69], which infer discrete subtype-specific stage-wise sequences of biomarker abnormalities. Importantly, while many studies have focused on single-domain characterizations (e.g., cortical atrophy only), the approach used here considers progression and staging as a multi-domain process (atrophy, biomarkers, cognition), enabling a more comprehensive account of AD progression. Future studies might incorporate subtyping and continuous multi-domain approaches for the study of functional alterations in AD. Finally, future research may provide insights into the mechanistic underpinnings of the results found in this study.

In conclusion, this study informed an extended ATN model incorporating EM network dysfunction. We found that over the course

of the disease, activation and deactivation during successful encoding diminish, partly as a consequence of tau pathology and partly as a consequence of neurodegeneration. These findings help to narrow a knowledge gap of the relationship between AD biomarkers, brain dysfunction and cognition.

## Methods

### Participants

Participants were part of the multi-center DZNE Longitudinal Cognitive Impairment and Dementia study (DELCODE)[37]. All participants gave written informed consent in accordance with the Declaration of Helsinki, and the DELCODE study was approved by the ethics committees of all participating institutions. The process was led and coordinated by the ethical committee of the medical faculty of the University of Bonn (trial registration number 117/13). All relevant ethical regulations were complied with. DELCODE was registered with the German Clinical Trials Register (https://www.bfarm.de/EN/BfArM/Tasks/German-Clinical-Trials-Register/_node.html; accession number: DRKS00007966).

In total, 1011 individuals were enrolled in the study. They participated in a range of neuropsychological tests, structural MRI and fMRI (resting-state and/or task-based, diffusion MRI) sessions, as well as cerebrospinal fluid (CSF) biomarker measurements. Groups were diagnosed based on criteria reported in ref. 37. The diagnosis of SCD was given if participants reported subjective memory impairment but scored above the −1.5 standard deviations (SD) cutoff on all subtests of the CERAD-plus test battery adjusted for age, sex, and education. Healthy subjects with above cutoff performance and without any subjective memory impairment were classified as CN. Participants with age-, sex-, and education-adjusted memory performance below −1.5 SD on the delayed recall trial of the CERAD-plus EM test were diagnosed with amnestic MCI. Participants were classified as DAT if they fulfilled NINDCS/ADRDA criteria and had a CERAD-plus score of below −1.5 SD, an extended MMSE score between 18 and 26, and a clinical dementia rating greater or equal to one. Additionally, first-degree relatives of patients with a diagnosis of suspected AD (ADrel) were included in DELCODE if their performance was within 1.5 SD in the CERAD-plus test battery.

Among the 1011 participants, we excluded 12 participants who converted to non-amnestic MCI by April 2021, and 23 additional subjects who did not belong to any of the AD-related biomarker groups (see section "CSF-biomarkers"). Finally, the group of ADrel was excluded from our analysis, since we did not have any a priori hypotheses regarding this subsample (for fMRI activation patterns in this group, see refs. 26,27). The final analysis sample ($n = 493$) consisted of 165 cognitively healthy controls (CN), 214 participants with SCD, 82 participants with MCI and 32 participants with DAT, all of whom had available baseline fMRI data. Among these, 218 (44%) participants had at least one available CSF measurement. Age ranged from 60 to 87 years at baseline (mean = 70.58, sd = 5.9, range = 27). Sex was self-reported, and the distributions were 257 females and 236 males in the analyzed sample. Disaggregated sex and gender data have not been collected. They were not considered in the study design.

### Cognition

As cognitive variables, we included the Preclinical Alzheimer's Cognitive Composite (PACC5)[70] score and the summary score from the cognitive subscale of the Alzheimer's disease assessment scale (version 13; ADAS-COG-13)[71]. The PACC5 score is a cognitive composite score reflecting various domains from delayed free and cued recall, results from the Mini-Mental State Examination[72]. The ADAS-COG-13 includes 11 items with tasks related to word recall and recognition, object naming, following commands, constructional and ideational praxis, orientation and comprehension[71].

### CSF-biomarkers

CSF biomarkers were obtained via lumbar puncture performed by trained study physicians. CSF samples were centrifuged, aliquoted, and stored at −80 °C for retests, and analyzed with commercially available kits (V-PLEX Aβ Peptide Panel 1 (6E10) Kit (K15200E); Innotest Phospho Tau(181P), 81581, Fujirebio Germany GmbH, Hannover, Germany). CSF biomarker data were available for 218 participants.

For this study, we obtained the Aβ42/40 ratio, pTau$_{181}$ and total tau (t-tau) values. Those were used (1) to train the DPM (see below) and (2) to establish AD-biomarker groups based on the AT(N) criteria[11]. For this, we used DELCODE-specific cutoff points determined via Gaussian mixture modeling from baseline data in a previous study[73] (amyloid-negative (A-): Aβ42/40 > 0.08; Amyloid-positive (A+): Aβ42/40 ≤ 0.08; Tau-negative (T-): $pTau_{181} < 73.65$; Tau-positive (T+): $pTau_{181} \geq 73.65$). More details on the procedures can be found in ref. 37.

### (f)MRI acquisition

For structural MRI, whole-brain T1-weighted magnetization prepared rapid gradient echo sequence (MPRAGE) images were acquired (TE = 437 ms, TR = 2500 ms, 7° flip angle, 1 mm isotropic) on 3 Tesla Siemens MRI scanner systems. Additionally, coronal T2-weighted turbo spin-echo images of the MTL and hippocampus (TE = 354 ms, TR = 3500 ms, 120° flip angle, 0.5 × 0.5 × 1.5 mm resolution) were obtained. For WMH, the MPRAGE images and T2-weighted Fluid-Attenuated Inversion Recovery (FLAIR) images were utilized (TE = 394 ms, TR = 5000 ms, full head coverage, 1 mm isotropic). Finally, fMRI was acquired by means of echo planar images (TE = 30 ms, TR = 2580 ms, 80° flip angle, 3.5 mm isotropic).

### fMRI task and task performance

During the fMRI session, participants were presented with a modified version of an incidental encoding task initially described by ref. 25 (for details on the modified version, see refs. 8,28,74). Participants were presented with 88 novel scenes, evenly split between scenes indoors and outdoors (44 each). Additionally, 44 repetitions of two pre-familiarized scenes—one indoor and one outdoor scene—were presented, each repeated 22 times. The task was administered using Presentation (Neurobehavioral Systems Inc). Participants had to classify each scene as either indoor or outdoor via a button press. Presentation time for each scene was set at 2500 ms. Following a break of 60 min, participants underwent a memory test in which they were instructed to rate their confidence in their recognition of the previously presented novel scenes using a 5-point Likert scale. From this recognition test, the area under the receiver operating characteristic curve of hits (i.e., correctly classified novel images as novel) versus false alarms (i.e., falsely classified familiar images as novel) was calculated (A′ – "A prime")[38]: $A\prime = \int_0^1 H(FA)dFA$, with H and FA being defined as Hits or False Alarms, respectively.

### fMRI preprocessing and first-level modelling

Functional MRI preprocessing was done in SPM12 (Welcome Trust Centre for Human Neuroimaging, UCL, UK). The functional MRIs were slice-time corrected, unwarped, realigned, segmented, coregistered with the structural images, normalized to a population standard space via geodesic shooting, normalized to MNI space via an affine transformation, and finally, spatially smoothed with a Gaussian filter kernel at 6 mm full width half maximum.

For the first-level general linear models (GLMs), the canonical hemodynamic response function with a 128-s high-pass filter and no global scaling was used (see ref. 28). Regressors of interest were the novelty regressor (all novel images), a successful memory regressor obtained from a parametric arcsine transformation of the novelty

regressor modulated with the confidence ratings from the recognition task[26], and a regressor for the familiar images. Additionally, each GLM included the six rigid-body motion parameters determined from realignment as covariates of no interest, plus a single constant representing the implicit baseline. The contrast of interest was the successful memory encoding contrast. It is based on the single onset regressor containing all novel scenes parametrically modulated by the confidence ratings x from the recognition task post-fMRI ($arcsine(\frac{x-3}{2} \times \frac{2}{\pi})$). Thus, the positive t-contrast we used for this study shows regions which are more strongly activated (positive numbers) and deactivated (negative numbers) for successfully encoded items. The choice for the parametric modulator was motivated by a recent study performing a model comparison identifying the parametric modulator as an improved characterization of encoding-related activations in old age in comparison to the standard categorical alternative of this contrast[74].

### Disease progression model for staging based on ATN pathology

For quantification of each patient's disease stage during progression towards AD, we utilized a continuous multivariate DPM based on Bayesian Gaussian Process (GP) regression[14,75] (code available from https://gitlab.inria.fr/epione/GP_progression_model_V2). In this approach introduced by ref. 14, GPs were used to empirically describe smooth monotonically increasing transition curves from normal to abnormal biomarker levels as proposed in Jack et al.[47] (see ref. [14] for mathematical details). The model can be described as follows:

$$y^j(t) = \left( y^j_{b_1}(t), y^j_{b_2}(t), \ldots, y^j_{b_{N_b}}(t) \right)^\top = \boldsymbol{f}(t) + \boldsymbol{v^j}(t) + \epsilon, \quad (1)$$

where $y^j(t)$ are the longitudinal biomarker measurements at time point $t$, which can be represented by the fixed-effect model $f \sim GP(0, \sum_G)$, the individual random effects $\boldsymbol{v^j}(t) = (\boldsymbol{v}^j_{b_1}(t), \boldsymbol{v}^j_{b_2}(t), \ldots, \boldsymbol{v}^j_{b_{N_b}}(t))^T$ and $\epsilon = (\epsilon_{b_1}, \epsilon_{b_2}, \ldots \epsilon_{b_{N_b}})^T$ observational residuals.

After time-reparametrization, the biomarker evolution over reparameterized disease time can be described as

$$y^j\left(\phi^j(\tau)\right) = \boldsymbol{f}\left(\phi^j(\tau)\right) + \boldsymbol{v^j}\left(\phi^j(\tau)\right) + \epsilon, \quad (2)$$

where $\phi^j(\tau) = \tau + d^j$. In this model, $d^j$ corresponds to the individual time-shift of a participant. For further mathematical details, please resort to ref. 14. Our DPM application in this study focused on the following groups of variables based on the ATN pathological classification system and cognitive performance: (A) Biomarkers from CSF were Aβ42/40 ratio and pTau$_{181}$ levels. (B) Atrophy measures comprised bilateral hippocampus and entorhinal cortex volumes from FreeSurfer (version 7, longitudinal pipeline, corrected for total intracranial volume (TICV)). (C) Cognition was operationalized in terms of the PACC5 and the ADAS-COG-13 summary scores. Importantly, the model inversion based on observation of those features (including available follow-ups) estimates the most probable individual patient's disease stage continuously (including its uncertainty) and a data-driven disease progression curve for each of the fitted biomarkers. In detail, according to ref. 14, the estimated disease stage is given as the expected value of

$$p(t^* | y^*, y, t, m, t') = \frac{(p(y^* | t^*, y, t, m, t') * p(t^*))}{(p(y^* | y, t, m, t'))} \quad (3)$$

### Participant selection for the model fit

For DPM fit, we only included DELCODE participants who had at least one complete measurement occasion of biomarkers[14]. Out of those, we excluded participants with a biomarker profile outside the Alzheimer's continuum (e.g., A-T+; see ref. 76). This was

further refined by restricting the clinical AD-risk groups (i.e., SCD, MCI, and DAT) to those with amyloid positivity (A+). Additionally, we used clinical conversion data to exclude participants who later converted to non-amnestic MCI or to non-Alzheimer's type dementia, such as dementia with Lewy bodies, Parkinson's disease dementia, semantic dementia, or vascular dementia. This resulted in a subsample of 208 participants (80 CN, 57A+ SCD, 44 amnestic A+ MCI, 27A+ DAT). Further details on the descriptive statistics of the model estimation sample and parameter settings, as well as schematic representation, can be found in the Supplementary Table 2 and Supplementary Fig. 6, respectively. After model fitting, model inversion was done on the remaining participants who were not part of the fitting process.

### White matter hyperintensities and effective connectivity

In our analysis of potential determinants of our results, we also included total white matter hyperintensity volumes using estimates from a previous study[77]. WMH Volumes were segmented using the AI-augmented version of the Lesion Segmentation Toolbox and both T1-weighted MPRAGE and T2-weighted FLAIR images.

For our effective connectivity metric, we used the effective connectivity estimate from the right parahippocampal place area (PPA) to the right precuneus (PCU) from a previous DELCODE study using DCM[44]. This study modeled our task-fMRI data and investigated the influences of CSF biomarkers on task-based effective connectivity of a-priori specified regions of interest. Extracted time-series data were corrected for age, sex, and years of education. Results showed that the effective connectivity from the right PPA onto the right PCU was modulated by the interaction between CSF biomarkers for amyloid and tau. Further details on the modeling approach and results can be found in ref. 44.

### Statistical analyses

All statistical analyzes (except for image-based analyzes) were carried out in R (version 4.2.1). A list of relevant packages used in this study can be found in Supplementary Table 11. Significance levels were set at $p < 0.05$. First, group differences in demographics were analyzed by one-way analyzes of variance. Wherever normality was violated, non-parametric Kruskal–Wallis tests were used. Group differences in sex were analyzed by means of chi-square tests. *Post-hoc* group differences were analyzed by two-sample *t*-tests or Wilcoxon rank-sum tests and corrected for multiple testing by using the Bonferroni–Holm correction. Relationships between disease stage and fMRI task performance were analyzed with bivariate partial correlations adjusting the variables for age, sex, education, and scanning site. Individual associations between biomarkers and disease stage were analyzed using Pearson and Spearman correlations. Next, to assess the association between disease stages and diagnostic groups and AT biomarker categories, respectively, we calculated separate one-way analyzes of covariance (ANCOVAs) with disease stage as the dependent variable and diagnostic group and AT biomarker status as independent variables, adjusting for age, sex, education and MRI scanning site.

After establishing a score of disease progression for AD, we were interested in the association of that progression score with the fMRI BOLD signal contrasts of memory encoding on the voxel-level. We expected a region-specific association of activity with disease progression. The analysis was carried out using the Sandwich estimator toolbox[78] (v2.2.2, MATLAB). All available longitudinal scans were included in the fitting process to obtain between-subjects effects and additionally estimate longitudinal effects of time. The design matrix included an intercept, linear and quadratic terms of the disease stage, intraindividual time coded as difference from the intraindividual mean timepoint in years, age, sex, education, and the interaction terms time*covariates. For inference, we used the Wild Bootstrap method[79] with 1000 repetitions. T-contrasts for the effects of interest (linear and

quadratic effects of disease stage) were obtained and thresholded using FWE correction with $p < 0.05$.

Next, we next were interested in the progression patterns of activation and deactivation over the disease stages. To do so, we extracted mean activation and deactivation values from the FWE-thresholded (de-)activation maps for each of the available measurement time points. Task-related activations and deactivations were analyzed separately, as they reflect functionally distinct processes: activations typically involve task-relevant networks, whereas deactivations are often associated with the default mode network and disengagement from internally directed cognition[19,80]. Treating them separately avoids averaging out potentially opposing effects and enables a more precise interpretation of BOLD signal changes[81–83]. These were submitted to smoothing splines using the ss() function from the *npreg* package in R[84]. The independent variable was the disease stage obtained from DPM. The dependent variable was the average activation or deactivation value. In addition to the function's default values, number of knots were a-priori set to 3. To estimate the time point of fastest activation change over the disease stages, we additionally calculated first derivatives of the cluster-level smoothing splines over disease stage using the *pracma* package in R[85]. This provided us with an estimate of change of EM-related deactivation and activation at each point along the disease trajectory line. This was done to compare the relative change magnitudes of markers from the DPM to the change magnitudes of EM activation.

Next, we were interested in the respective contributions of DPM markers to activity differences. For this, we first calculated two separate one-way ANCOVAs for the association between EM deactivation and activation and AT staging groups while controlling for hippocampal and entorhinal volume and demographics age, sex, and education. Second, we were interested in the association between volumes and activity while controlling for CSF biomarkers. For this, we calculated semi-partial Pearson correlations between EM activity and volumes adjusted for demographics and CSF-A$\beta$42/40 ratio and CSF-pTau$_{181}$. Finally, to check the association between activity and cognition, we calculated semi-partial Pearson correlations adjusting cognition for demographics and hippocampal and entorhinal volume.

Lastly, we aimed at testing above findings longitudinally. First, we calculated linear mixed effect models (LMEs) in R with EM activity as the dependent variable. The effect of interest was the interaction between time and continuous disease stage. Covariates were the interactions between time and demographics age, sex and education. We calculated LMEs with and without time as random slope effect, but the former had singular fits, so the latter ones were used for the analysis. Second, we were interested if longitudinal activation change over follow-ups would align with the actual intra-individual stage progress of an individual over follow-ups (as indicated by ATN and Cog). For this, we obtained disease stage estimates for each measurement occasion independently for each participant using the available data. Next, we used linear regression to estimate rates of change along the disease progression with longitudinal disease stage estimates as the dependent variable and intra-individual time as the independent variable. Additionally, we repeated this for longitudinal mean activation values by submitting the activity values as the dependent variable and intra-individual time as the independent variable to simple linear regressions. Finally, we extracted the beta coefficients from both simple linear regressions and correlated them using Pearson correlation. We expected that higher change in contrast value would be related to a faster disease progression.

### Reporting summary
Further information on research design is available in the Nature Portfolio Reporting Summary linked to this article.

## Data availability
The raw data collected in the study "DELCODE−DZNE-Longitudinal Cognitive Impairment and Dementia Study (BN012)" cannot be made openly available without violation of the data protection concept of the DZNE. The same applies to the processed individual (f)MRI images. Access to the relevant study data can be obtained by submitting an application to the Clinical Research Platform of the DZNE. The template for the application for the submission of data and biomaterial samples is available on the DZNE homepage (https://www.dzne.de/en/research/research-areas/clinical-research/databases-of-the-clinical-research/). The expected timeframe for response to access requests is 1 month. Access will be granted for 10 years. Source data are provided with this paper.

## Code availability
Code to the DPM utilized in this study and for the visualization of Figs. 1a and 1b can be found under https://gitlab.inria.fr/epione/GP_progression_model_V2. Code for the second-level GLM and the parallel mediation model, and the surface renderings can be found on GitHub at https://github.com/renelattmann/EMN_dysfunction[86].

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

## Acknowledgements

The study was funded by the German Center for Neurodegenerative Diseases (Deutsches Zentrum für Neurodegenerative Erkrankungen (DZNE)), reference number BN012. This work was supported by the German Research Foundation (Deutsche Forschungsgemeinschaft, DFG; Project ID 374011584/3T Ganzkörper MR-Tomograf, to P.D.; Project ID 425899996 – CRC 1436).

## Author contributions

R.L., G.Z., and E.D. conceptualized the present data analysis. Overall design, implementation and collection of data for the DELCODE study at the different study sites was provided by E.I.I., H.S., O.P., J.H.-R., D.Gr., M.B., F.L., M.C., A.S., S.S., L.S.S., J.P., E.J.S., S.A., K.F., J.W., A.R., A.Ro., W.G., S.T., I.K., D.Go., C.L., P.D., S.H., K.S., B.H.S., F.J., and E.D. L.K., E.I.I., J.B., R.Y., A.Sp., M.Sch., M.St., F.B., M.W., and E.D. provided core methodological data. R.L., J.M., E.I.I., J.B., R.Y., and G.Z. preprocessed the MRI data. R.L. performed the analyses and visualizations. R.L. wrote the initial draft of the manuscript. J.M., N.V., L.K., E.I.I., R.Y., M.W., Y.S., B.H.S., E.D., A.M., and G.Z. reviewed and edited the manuscript. G.Z. and E.D. supervised the present project equally.

## Funding

## Competing interests

E.D. reports personal fees from Biogen, Roche, Lilly, Eisai and UCL Consultancy as well as non-financial support from Rox Health. He is scientific co-founder of neotiv GmbH and owns company shares. All other authors declare no competing interests.

## Additional information

[1]German Center for Neurodegenerative Diseases (DZNE), Magdeburg, Germany. [2]Institute of Cognitive Neurology and Dementia Research, Otto von Guericke University Magdeburg, Magdeburg, Germany. [3]Department of Neuroimaging Sciences, Institute for Neuroscience and Cardiovascular Research, Row Fogo Centre for Research into Ageing and the Brain, University of Edinburgh, Edinburgh, UK. [4]Department of Artificial Intelligence in Biomedical Engineering (AIBE), Friedrich-Alexander Universität Erlangen-Nürnberg (FAU), Erlangen, Germany. [5]Department for Psychiatry and Psychotherapy, University Clinic Magdeburg, Magdeburg, Germany. [6]German Center for Neurodegenerative Diseases (DZNE), Bonn, Germany. [7]Institute for Medical Biometry, Informatics and Epidemiology, University Hospital Bonn, University of Bonn, Bonn, Germany. [8]Department for Old Age Psychiatry and Cognitive Disorders, University Hospital Bonn, Bonn, Germany. [9]Department of Neurology, University Hospital Bonn, Bonn, Germany. [10]MR-Research in Neurosciences, Department of Cognitive Neurology, University Medical Center Göttingen, Göttingen, Germany. [11]Department for Biomedical Magnetic Resonance, University of Tübingen, Tübingen, Germany. [12]Charité – Universitätsmedizin Berlin, Berlin Center for Advanced Neuroimaging, Berlin, Germany. [13]Cologne Excellence Cluster on Cellular Stress Responses in Aging-Associated Disease (CECAD), University of Cologne, Cologne, Germany. [14]Department of Psychiatry and Psychotherapy, Division of Neurogenetics and Molecular Psychiatry, Faculty of Medicine and University Hospital Cologne, University of Cologne, Cologne, Germany. [15]Department of Psychiatry & Glenn Biggs Institute for Alzheimer's and Neurodegenerative Diseases, San Antonio, TX, USA. [16]German Center for Neurodegenerative Diseases (DZNE), Tübingen, Germany. [17]Section for Dementia Research, Hertie Institute for Clinical Brain Research and Department of Psychiatry and Psychotherapy, University of Tübingen, Tübingen, Germany. [18]Department of Psychiatry and Psychotherapy, University of Tübingen, Tübingen, Germany. [19]German Center for Neurodegenerative Diseases (DZNE), Berlin, Germany. [20]Department of Psychiatry and Psychotherapy, Charité – University Medical Center Berlin, Berlin, Germany. [21]Institute of Psychiatry and Psychotherapy, Charité – University Medical Center Berlin, Berlin, Germany. [22]German Center for Neurodegenerative Diseases (DZNE), Göttingen, Germany. [23]Department of Psychiatry and Psychotherapy, University Medical Center Göttingen, University of Göttingen, Göttingen, Germany. [24]Leibniz Institute for Neurobiology, Magdeburg, Germany. [25]Department of Medical Sciences, Neurosciences and Signaling Group, Institute of Biomedicine (iBiMED), University of Aveiro, Aveiro, Portugal. [26]German Center for Neurodegenerative Diseases (DZNE), Rostock, Germany. [27]Department of Psychosomatic Medicine, Rostock University Medical Center, Rostock, Germany. [28]Department of Psychiatry, Medical Faculty, University of Cologne, Cologne, Germany. [29]Department of Psychiatry and Psychotherapy, School of Medicine, Technical University of Munich, Munich, Germany. [30]University of Edinburgh and UK DRI, Edinburgh, UK. [31]Department of Psychiatry and Neurosciences, Charité – University Medical Center Berlin, Berlin, Germany. [32]ECRC Experimental and Clinical Research Center, Charité – University Medical Center Berlin, Berlin, Germany. [33]Institute for Biology, Otto von Guericke University Magdeburg, Magdeburg, Germany. [34]Institute of Cognitive Neuroscience, University College London, London, UK. [35]These authors contributed equally: Gabriel Ziegler, Emrah Düzel. ✉e-mail: rene.lattmann@dzne.de

