## [Transparent Peer Review file · Nature Communications]

Dysfunction of the episodic memory network in the Alzheimer's disease cascade

Corresponding Author: Mr René Lattmann

Version 0:

Reviewer comments:

Reviewer #1

(Remarks to the Author)

In this study, Lattman and colleagues investigate episodic memory circuit function as measured with task fMRI as a potential marker of AD-related cognitive decline prior to the neurodegeneration typically considered the driver of this cognitive impairment. In a sample of 208 individuals from the DELCODE study, they first derive a continuous disease stage measure from longitudinal CSF biomarker, MTL regional volume, and cognitive data. In a larger sample with longitudinal fMRI data, they then show that regions of task-related activation and deactivation show abnormality associated with the derived disease stage measure, and present evidence that these task fMRI changes correlate with cognitive decline and are independent of atrophy. The authors characterize their findings as evidence of synaptic dysfunction along the AD cascade that is independent of neurodegeneration.

Overall, the study provides interesting insight into the neural correlates of cognitive decline and utilizes a somewhat unique longitudinal dataset to explore the independent and combined influence of different factors along the AD pathway on cognitive decline. The manuscript is well-written with nicely designed figures, but I have a few concerns related to the clarity of the described methods and the authors' interpretation of their results that should be addressed prior to publication.

Major concerns:

- The major claim of the paper seems to be that AD-related fMRI changes occur independent of neurodegeneration, but the data presented are not fully convincing. The disease-related deactivation and activation shown in Figures 2C-D show increases in biomarker abnormality prior to increases in CSF amyloid/tau, so can it really be claimed that these are circuit changes associated with the AD pathway and not aging-related changes independent of AD? In addition, the authors show the relationship between fMRI activity and cognition remains after adjusted for MTL regional atrophy, but what about directly testing this with a mediation model rather than simply adjusting for regional volume?
- Related to the point above, in Figure 4 the authors show that deactivation is most prominent at later disease stages i.e. increases for individuals above the median split of their continuous disease stage measure, and decreases for those below the median split for this measure. Though critical for their conclusions, this finding met the threshold for significance for deactivation but not activation, and was based on a median-split approach that was not justified. Can the authors explain why they dichotomized their continuous measure only for this analysis?
- The disease progress model is an interesting approach but raises some questions that should be answered. What makes this approach Bayesian, is it the prior of a monotonic sigmoidal that the data must be fit along? How does this approach differ from other population trajectory modeling approaches like sampled iterative local approximation (SILA) and why was the DPM approach chosen for this study? Is it appropriate to include participants with just one time point in the DPM approach (as seen in Figure S1) for fitting a longitudinal trajectory? Also, please include the model formula in the methods or supplement.
- Though the manuscript reads quite nicely, much important methods info is omitted that would be confusing for a reader going through the manuscript sequentially, without having to jump to the methods at the end. For example, associations with fMRI task performance are reported at the end of Section 2.1, but the task itself has not been described, nor how the A' measure was computed or what it indicates. It also not clear what exact data is being used for each analysis stage, e.g. how many fMRI time points are used to generate the activation trajectories in 2C-D. In some cases this information is in the figure legend but should be in the manuscript text, please check to make sure important methods info is not missing.

Minor concerns:

- Tau positivity of the ATN framework was defined using ptau_{181} , but this measure tends to be less predictive of tau pathology than ptau_{217} , and tends to represent pathology change somewhere after amyloid accumulation but before significant tau pathology. Can the authors justify using this measure to define tau positivity?
- Was the sample used in the DPM analysis 208 individuals as stated in the text, or 210 as stated in the legend for Figure 1?
- Why do the authors think sex was not related to more advanced disease stage, given the growing body of work suggesting that women accumulate AD pathology more quickly than men?
- Do the density plots in Figure 1C and 1D only show baseline data?
- What is the p-value threshold for the voxelwise results displayed in 2B (not stated in text, hard to tell from legend)?
- Are the results described in section 2.4 adjusted for age, sex, and education? If not, could most of these results be explained by an association with age?
- For clarity, figure 4B should show the same trajectories as 4A i.e. with the early and late stage groups separated, even if the result is not significant.
- In view of less pronounced activity differences between successful encoding with declining cognition, the authors should mention the literature around network dedifferentiation in aging and AD in the discussion section.
- In the discussion section Fig. 2C is said to show fMRI deactivations becoming abnormal after CSF and MTL volume change, but the orange line appears to increase prior to any of these other variables?

Reviewer #2

(Remarks to the Author)

This study presents an ambitious and well-structured investigation into episodic memory network dysfunction in the aging - Alzheimer's disease (AD) continuum, using an extensive and enviable dataset from the DELCODE study. The authors integrate task-based fMRI – as a proxy of synaptic dysfunction - within the amyloid-cascade disease progression model (DPM). The authors showed widespread loss of deactivation and activation with disease progression which related with cognitive decline (partially independent of neurodegeneration – as quantified by regional atrophy). The authors conclude that synaptic dysfunction and neurodegeneration are independent drivers of cognitive decline. The methods are generally state-of-the-art, and the conceptual framework is generally clear. I have however several conceptual and methodological concerns that I think require further clarification.

Major comments

1) Operational circularity in defining synaptic dysfunction. The study defines synaptic dysfunction in terms of the fMRI signal changes that are being measured. This creates a risk of conflating the measurement (i.e., fMRI signal) with the underlying physiological construct of synaptic dysfunction. Interindividual fMRI variability in the context of aging and disease can reflect multiple factors—such as individual cognitive strategies, neurodegeneration (which is partly controlled for), vascular changes, and other noise sources. The assumption that fMRI-activity reflects synaptic dysfunction is central to the manuscript's claims appears either questionable or circular. The interpretation of the findings may therefore be a bit forced and oversimplistic .

2) Choice of Parametric Modulator for Signal Variability. The authors employ a parametric modulator of successful encoding as a proxy for synaptic dysfunction. It is unclear why this specific measure was selected over other potential fMRI metrics. Since AD participants tend to successfully encode fewer items, regression towards the mean could partially account for the observed results. Moreover, increased noise in AD individuals might also explain the relationship between task-fMRI metrics and the AD continuum. Relatedly (and very minor comment; out of curiosity), activation and deactivation measures have been combined in the past. Why are now reported separately? Is there any theoretical ground; if so, please explain.

Minor comments

3) Clarity in Presenting Data and Task Description. The presentation of the results is somewhat difficult to follow because the details of the data and the specifics of the fMRI task are not introduced early in the manuscript. Providing a more thorough early description of the experimental design and the fMRI task would improve the reader's understanding. Additionally, effect sizes are inconsistently reported; including them consistently would enhance the interpretability of the results.

4) While the DPM model uses longitudinal data; the fitting is highly dependent on cross-sectional information. I think this probably needs to be acknowledged. This might be harsh, as the DPM models have mostly relied on cross-sectional data. yet the results are highly dependent on the relationship between “cross-sectional” and “longitudinal” distributions + reliability of the measure. Think of memory or hippocampal volume in towards the healthy spectrum: the large variability in cross-sectional data can obscure real changes. The small variability cross-sectional in some biomarkers at baseline levels imply that small changes will be detectable quite early.

5) The DPM model and the later correlations with disease-ages are performed in partially overlapping samples. Please, clarify when and if there is some circularity in the analyses. I might have missed it; but are the activation and deactivation regions constrained by the Disease Stage Association model?

Reviewer #3

(Remarks to the Author)

This study examines the relationship between brain function and markers of Alzheimer's disease in 493 participants from the DELCODE study. Using CSF measures of amyloid and tau, along with measures of hippocampal and entorhinal volume, the results show that brain activity is most closely associated with hippocampal volume and tau levels. Brain activity was also associated with a baseline measure of disease stage which represented a compilation of the various markers. The study is well thought out and the analyses are in depth. I particularly appreciate the examination of brain function in relation to the markers of AD, as the symptomology of dementia all comes down to how well the brain continues to work in the face of

developing neuropathology. I have the following comments.

The disease stage is set on an arbitrary scale of years. Is there a way to anchor the disease stage to the diagnosis year in those with cognitive impairment?

Figure 1 and other places. The term 'model posterior' should be more well defined.

2.5 Longitudinal activation change is moderated by disease stage. It should be clear that the association between longitudinal change in activation was observed with baseline disease stage. It is also interesting that the baseline analyses shows a relationship, but no relationship is seen with longitudinal disease state change. Why would that be?

Information is needed on the longitudinal data. For example, what is the mean interval of the various longitudinal assessments (CSF, fMR, etc)? This should be added to the text in appropriate places.

4.6 fMRI preprocessing and first-level modelling. More info is needed to explain the exact contrast assessed in the analyses. For example, here it says 'the successful memory encoding contrast' was used. What was the baseline for this contrast? The discussion states the differences between remembered and forgotten stimuli, but that is unclear here.

Discussion, paragraph 2. It would be helpful to reiterate what the time points of neuropathologic change are.

Reviewer #4

(Remarks to the Author)

Lattman et al. present a novel application of disease progression modeling to estimate Alzheimer's disease progression stages. The estimated disease stages were then related to longitudinal changes to episodic memory ability as well as localized changes in functional activation/deactivation during memory encoding. This capacity of the methods presented in this work to facilitate new insights into the progression of Alzheimer's disease is evident, however, the manuscript also demonstrated some weaknesses that prevent me from recommending it for publication in its current state.

My primary concerns deal with the presentation of the author's methods, and with some theoretical weaknesses presented in the introduction and/or discussion sections of the paper. Within that first domain, I find the author's presentation of their DPM method to be somewhat opaque to readers not already familiar with that methodology and request that the DPM methodology is more clearly conveyed throughout the manuscript (see comments 3, 16, and 21 below). Also pertaining to methodology, the author's classification of "disease stage" – which many of their analyses and the interpretation of their results rely on – is not made entirely clear in this manuscript (see comments 8-10 and 15). Finally, I feel that the authors' conceptualization of "synaptic dysfunction" makes some unwarranted assumptions about the mechanisms underlying Alzheimer's disease pathology, which should be qualified or corrected (see comments 6, 7, 17-19).

Specific Comments:

1. Abstract.

a. "loss of deactivation" juxtapose with "hyperactivation" (meaning increase in activation) is confusing, if technically correct. Please re-word for clarity.

2. p3 "western societies".

a. Why specify western societies here? Surely there is enough evidence to conclude that Alzheimer's disease is of global concern.

3. p3 "disease progression model (DPM)".

a. The methodological advancement of this paper hinges on this statistical approach - more effort should be made to describe this modeling approach - how does DPM differ from other longitudinal modeling approaches that have been applied to Alzheimer's disease progression in the past?

4. p3 "longitudinal CSF biomarkers"

a. Example of such biomarkers would be useful

5. p3 "It is one of the first cognitive faculties"

a. Ambiguous "It" here - initially unclear if authors are referring to episodic memory or default mode network function.

6. p3 "The prevailing cascade models of AD in humans are not compatible with this possibility because they consider memory impairment to be a consequence of neurodegeneration rather than synaptic dysfunction"

a. This is not entirely true - cognitive and neurological scaffolding mechanisms allow for the maintenance or repletion of cognitive ability in light of pathological decline, though any gains are likely to be short-lived in cases where neurodegenerative disease is present. See the Park & Reuter-Lorenz STAC model series of papers.

7. p4 "We defined synaptic dysfunction as a brain activity abnormality that cannot be explained by neurodegeneration"

a. Why "synaptic" in that case? This definition seems to indicate functional changes that are not paired with pathological structural changes... cardiovascular mechanisms and functional connectivity mechanisms are just as likely as mechanisms

operating within the synapse. What is the rationale behind emphasizing synaptic dysfunction rather than these other factors or a combination of these factors?

8. p5 "Second, each participant was assigned a continuous disease stage value... Bivariate correlations between DPM markers and obtained disease stages are provided in Supplementary Figure 2."

a. A brief summary of these disease stages would be appropriate here, especially considering the Methods section of this paper is online-only. Was this classification based on Braak's staging for disease progression? Something else?

9. p5 "disease stage value on an arbitrary scale in years"

a. What does this mean? What does this arbitrary scale correspond to, and why was it equated to years?

10. p5 "Specifically, higher age was indicative of a more advanced disease stage... All further analyses with disease stage consider the age-, sex-, and education-corrected disease stage values."

a. Other demographic factors, particularly racial origin and socio-economic status, would be valuable to consider if those data are available.

11. p5 "As expected, participants further in disease progression showed worse A' scores of fMRI task performance... accounting for age, sex, and education."

a. What tasks were these, and more importantly what cognitive abilities did they correspond to?

12. p7 "we did not find indications for inverted U-shaped associations between disease stage and successful memory encoding."

a. Did you test for any other non-linearities aside from an inverted U function?

13. p7 "These later changes were accompanied by changes in hippocampal and entorhinal cortex volume."

a. How was this coupling statistically verified?

14. p9 "Random-intercept random-slope models did not converge, which is why we report on the random intercept models only."

a. How do you interpret this lack of convergence? Sample vs noise issue? Something else?

15. Figure 4 caption "The continuous disease stage scores were used to group participants into early and late stages based on the median disease stage (-3.37) in the sample."

a. What specifically do you define as "early" vs "late" disease stage

16. p10 "DPM"

a. A re-summation of this methodological approach at the beginning of the discussion section would aid in reader comprehension.

17. p11 "Thus, individuals in our sample show less pronounced brain activity differences between successfully memorized and later forgotten stimuli as cognition declines, an observation in line with the recently reported risk-dependent reduction of fMRI subsequent memory effects across the Alzheimer's risk spectrum"

a. The justification for this conclusion isn't immediately clear from the results presented - a reference to the specific findings which lead to this conclusion would be helpful.

18. p11 "episodic memory circuitry dysfunction"

a. What mechanism specifically does this phrase refer to?

19. p12 "episodic memory circuitry dysfunction"

a. Again, what does this refer to? It is unclear what you mean by "circuit" in this context.

20. Figure 5

a. This figure could be simplified by replacing the two unidirectional arrows connecting nodes N and E with a single bidirectional arrow.

b. Also, the left-justified placement of E vs N, implying a later place in the causal change, may be unwarranted considering the bidirectional influence of neurodegeneration and episodic memory function, and the direct effect of Tau on both processes.

21. p16-17, section 4.7 "Disease progression model for staging based on ATN pathology"

a. This section is crucial for understanding your findings. I encourage you to import as much of a summary of the DPM method as can be accommodated in the introduction/results section as is feasible

22. p17 "This resulted in a subsample of 208 participants"

a. The vast majority of your analysis are restricted to this smaller participant pool, no? If so, presenting your sample as 1,011 individuals is misleading

Reviewer comments:

Reviewer #1

(Remarks to the Author)

The authors have thoroughly responded to a variety of concerns raised by the (four!) reviewers, improving the clarity and rigor of their manuscript. I appreciate the response to all reviewer points both theoretical and technical, and believe the authors have contextualized their very interesting results in the literature. In my opinion, these changes have greatly improved the manuscript and it is suitable for publication.

Reviewer #2

(Remarks to the Author)

The authors have addressed all the comments thoroughly. I have no additional comments.

Reviewer #3

(Remarks to the Author)

The authors have addressed my comments and requests for further information.

Reviewer #4

(Remarks to the Author)

I thank the authors of this manuscript for their thoughtful and thorough responses to my comments on the previous submission. My concerns have been addressed and I am happy to recommend this version of the manuscript for publication.

Rebuttal Letter to reviewer comments on „Dysfunction of the episodic memory network in the Alzheimer’s disease cascade“.

R0.1: We thank the editor and reviewers for their in-depth constructive comments and for the helpful suggestions. In our revised manuscript, we have addressed all of these comments, incorporated new variables and data and conducted extensive additional analyses. We believe these serve to demonstrate the validity and specificity of our findings.

To summarize the changes described in the detailed responses below, we have expanded on descriptions regarding the disease progression modelling approach and deepened the method section dealing with the fMRI task and contrast design. Additionally, we have conducted new analyses targeting potential confounders of our activation results including vascular, connectivity, volumetric, residual variability and individual cognitive strategy contributions. In accordance, we made changes to the introduction and discussion providing more depth and breadth towards alternative explanations to our effects other than synaptic dysfunction.

We have revised the entire manuscript to provide greater clarity in relation to methods and results, and extended the discussion of potential mechanisms behind activation differences along the disease trajectory.

We trust that we successfully addressed all issues and that the revised manuscript can now be considered suitable for publication.

REVIEWER COMMENTS

Reviewer #1

In this study, Lattmann and colleagues investigate episodic memory circuit function as measured with task fMRI as a potential marker of AD-related cognitive decline prior to the neurodegeneration typically considered the driver of this cognitive impairment. In a sample of 208 individuals from the DELCODE study, they first derive a continuous disease stage measure from longitudinal CSF biomarker, MTL regional volume, and cognitive data. In a larger sample with longitudinal fMRI data, they then show that regions of task-related activation and deactivation show abnormality associated with the derived disease stage measure, and present evidence that these task fMRI changes correlate with cognitive decline and are independent of atrophy. The authors characterize their findings as evidence of synaptic dysfunction along the AD cascade that is independent of neurodegeneration.

Overall, the study provides interesting insight into the neural correlates of cognitive decline and utilizes a somewhat unique longitudinal dataset to explore the independent and combined influence of different factors along the AD pathway on cognitive decline. The manuscript is well-written with nicely designed figures, but I have a few concerns related to the clarity of the described methods and the authors' interpretation of their results that should be addressed prior to publication.

Major concerns:

R1.01: The major claim of the paper seems to be that AD-related fMRI changes occur independent of neurodegeneration, but the data presented are not fully convincing. The disease-related deactivation and activation shown in Figures 2C-D show increases in biomarker abnormality prior to increases in CSF amyloid/tau, so can it really be claimed that these are circuit changes associated with the AD pathway and not aging-related changes independent of AD?

Response to R1.01: We thank the reviewer for the important point. We agree with the reviewer that aging is a potential contributor to fMRI signal changes (Corriveau-Lecavalier et al., 2024; Maillet & Rajah, 2014; Kennedy et al., 2015; Soch et al., 2021). Additionally, AD biomarker accumulation confers a risk for hyperactivation in task-fMRI (for a review, see Corriveau-Lecavalier et al., 2024). However, whether age should be seen as playing the role of a confounder or as a causal risk factor that is likely to affect AD-pathology (ref xyz) and also affects BOLD changes (directly and indirectly) is a complex issue. For instance, even if in a hypothetical design, initially low-pathology subjects could be followed up from baseline, and might show increases of disease progression scores over time (assuming one waits long enough). These subjects would naturally increase with age. Consequently, we believe orthogonalizing aging and disease progression might be a pragmatic tool to study these effects specifically but is at the same time (1) likely to induce bias via overcorrection (due to multicollinearity by construction) or underadjustment (since also resting on a certain model assumption how age contributes to effects of interest). When focusing on figure 2C & D in the original manuscript, the depiction of the Disease Progression Model (DPM) overlaid with the fMRI-related smoothing spline results indeed show the activation curves before any correction for age. Since subjects included in the DPM do vary across age cross-sectionally, the provided progression curves might include contributions of age-related differences as a potential confounder. Being aware of the above concerns we have repeated the spline analysis of encoding-related activation and deactivation accounting for age differences via residualization (resting on a linearity assumption). In the supplemental figure below, we show the revised abnormality curves for fMRI task deactivations overlaid with the biomarker curves from the DPM (left) and the comparison of deactivation curves (right) for raw values (green) versus age-corrected values (blue). While the general pattern of disease progression for memory encoding-related deactivations

remained consistent even after the correction for age-differences, the initial increase in abnormality was found to be slightly attenuated after correction and is now more aligned with the PACC5 progression curve during the earliest disease stages. In alignment with the initial results, we similarly observed a steep increase in abnormality around disease stage of 5 years for the age corrected progression curve of brain deactivation. For the brain activation cluster, the results did not change, yet we included the fit for transparency. Thus, our results now show AD-related deactivation differences over disease progression that are unlikely to be affected by age. The results are in line with previous findings studying neural hyperactivation in the context of AD in-vitro (Busche et al., 2012) and in human participants (Leal et al., 2017), indicating a potential vicious cycle of hyperactivation and AD biomarker accumulation (Zott et al., 2019). In the revised version of the manuscript, we now report on the spline results adjusted for age. This includes an updated results section 2.3 with new time points of fastest changes for activity as well as new figures 2C and 2D that now show age-corrected activity abnormality curves. However, a complete dissection of aging and disease progression related effects is inherently complex, as disease progression and aging occur at the same time. To make this clear, we have now altered our limitations section that includes a discussion on the study of the correlated nature of disease progression and aging.

Limitations: “Seventh, aging represents the primary risk factor for AD, and advancing age is tightly coupled with both increased incidence and greater disease burden⁶⁶. Accordingly, a complete disentanglement of effects attributable to AD progression from those attributable to normative aging is not only challenging but may obscure meaningful sources of variance within patient samples. In the present study, we sought to mitigate this confounding by statistically residualizing for chronological age across all analyses. However, this approach assumes a linear relationship between age and neural measures, an assumption that may not fully capture the complex, potentially nonlinear interactions between aging and disease-related processes. A more precise characterization of these interactions was beyond the scope of the current manuscript but represents an important direction for future work.” (lines 485 – 494)

New Figure 2:

Figure 2. Association between encoding-related activity and disease stages. **A** Surface representation of the average activation (red colors) and deactivation (blue colors) for successful memory encoding extracted from the second-level SwE model ($n = 493$). **B** Linear associations of disease stage with successful memory encoding related activation (red colors) and deactivation (blue colors) from the second-level SwE model. Results were obtained using the Wild bootstrap method with 1000 repetitions thresholded at $p_{FWE} < 0.05$. **C** Relative position of biomarker abnormality differences for age-corrected encoding-related deactivation in relation to the biomarkers from our data-driven DPM approach ($n = 493$). FMRI abnormality starts in the earliest disease stages. Towards disease stage 0, a small plateau can be observed before increasing in abnormality together with volumetric biomarkers. **D** Relative position of biomarker abnormality change for the age-corrected successful memory encoding related activation. For activation, we only observed a linear increase in abnormality over the disease progression.

Supplementary Figure: Comparing effects of aging on curve estimation.

R1.02: In addition, the authors show the relationship between fMRI activity and cognition remains after adjusted for MTL regional atrophy, but what about directly testing this with a mediation model rather than simply adjusting for regional volume?

Response to R1.02: We thank the reviewer for this helpful suggestion. In response, we performed parallel mediation analyses to examine whether the relationship between disease-related memory encoding activity and cognition was mediated by hippocampal and entorhinal volumes combined using the lavaan package in R. Accompanying path diagrams can be found in the figure below. For our analyses, we controlled all variables for the demographics age, sex, and years of education a priori. Standard errors were bootstrapped based on 5000 resamples; model parameters were estimated using maximum likelihood. First, we investigated the association between deactivation and PACC5 (Figure Part A). Both hippocampal (indirect effect: $a*b = 0.056$, $SE = 0.016$, $z = 3.404$, $p = 0.001$) and entorhinal volume (indirect effect: $a*b = 0.020$, $SE = 0.010$, $z = 2.102$, $p = 0.036$) mediated the association deactivation and the PACC5. However, even though the mediators were accounted for by the model, the deactivation was significantly associated with the PACC5 (direct effect: $c' = 0.151$, $SE = 0.036$, $z = 4.172$, $p < 0.001$). Therefore, MTL volumes partially mediate the association between deactivation and the PACC5. Next, we investigated the parallel mediation of MTL volumes on the association between activation and ADAS-Cog-13 sum score. Here, hippocampal volume mediated the association between activation and the ADAS-Cog-13 sum score (indirect effect: $a*b = 0.068$, $SE = 0.018$, $z = 3.749$, $p < 0.001$). Entorhinal volume, on the other hand, did not contribute to the mediation (indirect effect: $a*b = 0.013$, $SE = 0.010$, $z = 1.377$, $p = 0.168$). Again, the deactivation was significantly related to cognition, even after accounting for the two MTL volumes (direct effect: $c' = 0.127$, $SE = 0.036$, $z = 3.490$, $p < 0.001$). Thus, these additional analyses do suggest that hippocampal volume is a partial mediator of the association between deactivation and the ADAS-Cog-13 sum score.

Mediation analysis of hippocampal volume and the association between fMRI-related deactivation and cognition. **A** Hippocampal volume significantly mediates the association between deactivation and the ADAS-Cog-13 sum score. **B** Additionally, hippocampal volume partially mediates the association between fMRI-related deactivation and the PACC5 score. ADAS13: ADAS-Cog-13 sum score; HC volume: Hippocampal volume. Note: The volume was multiplied by -1, such that larger values indicate higher pathology. All variables were corrected for age, sex, and years of education **a priori**. Sample size (PACC5): n = 468; Sample size (ADAS-Cog-13): 482. Different sample sizes result from missing data in each cognitive marker. $p < .01^{**}$; $p < .001^{***}$.

Second, we tested the potential mediation of MTL volumes on the association between activation and cognition (Figure parts C & D). Neither hippocampal volume (indirect effect: $a*b = -0.030$, $SE = 0.016$, $z = -1.897$, $p = 0.058$) nor entorhinal volume (indirect effect: $a*b = -0.016$, $SE = 0.008$, $z = -1.909$, $p = 0.056$) mediate the association between activation and the PACC5 (direct effect: $c' = -0.109$, $SE = 0.036$, $z = -3.054$, $p = 0.002$). Lastly, only hippocampal volume (indirect effect: $a*b = -0.038$, $SE = 0.017$, $z = -2.250$, $p = 0.024$), but not entorhinal volume (indirect effect: $a*b = -0.010$, $SE = 0.008$, $z = -1.273$, $p = 0.203$) was a partial mediator on the association between activation and the ADAS-Cog-13 sum score (direct effect: $c' = -0.118$, $SE = 0.038$, $z = 3.064$, $p = 0.002$). In summary, when including MTL volumes as mediators in a parallel mediation model, no full mediation was identified. Thus, the results were highly consistent with the original findings and we therefore retained the original results at that section in the manuscript. However, we leveraged the reviewer's suggestion in a separate analysis and response to reviewer 2 (Response to R2.01), which ultimately also resulted in a figure we included into the supplementary material. There, we analyze additional potential mediators such as cerebrovascular and effective connectivity metrics over and above MTL volumetry. Even when additionally incorporating the additional mediators into the model, the associations between progression-related memory-encoding activity and cognition remained significant. For a full description of this analysis, we would like to draw the reviewer's attention to our Response to R2.01. At this point, we would like to conclude that also a direct test for the influence of MTL volume did not change our results from the initial version of the manuscript, highlighting that neural activity is still related to cognitive performance showing a partial independence of activity and volume in their association with cognition. In response to the reviewer's helpful comment, we have added a section in the results.

Results: "This was also true for a direct test via mediation analyses." (lines 276)

Further results were explained upon in the response to R2.01, but we provide an excerpt of our answer here for completeness:

Results: “ In order to discern whether the associations survive the correction for additional third variables, we performed parallel mediation analyses including volume, white matter hyperintensities (WMH), and effective connectivity. Effective connectivity values were obtained from a recent study utilizing dynamic causal modeling⁴⁴. WMH were log-transformed and all variables entering the model were corrected for baseline age, sex, and years of education a priori. Supplementary Figure 5 provides an illustration of this mediation analysis. The complete model outputs including fit indices can be found in the supplementary tables 6 and 7. For disease-related activation, hippocampal volume (indirect path: $z = 2.102$, $p = 0.036$) and effective connectivity (indirect path: $z = 2.057$, $p = 0.040$) mediated the association between activation and cognition. Entorhinal volume (indirect path: $z = 1.580$, $p = 0.114$) and WMH (indirect path: $z = 1.023$, $p = 0.306$), on the other hand, did not significantly mediate the association. Even if mediators were taken into account, higher levels of cognition were related to higher activation (direct path: $z = 4.117$, $p < 0.001$). For disease-related deactivation (Supplementary Figure 4B), again, hippocampal volume (indirect path: $z = -2.211$, $p = 0.027$) and effective connectivity (indirect path: $z = -2.354$, $p = 0.019$) partially mediated the association between deactivation and cognition (direct path: $z = -4.799$; $p < 0.001$), but not WMH (indirect path: $z = -1.414$, $p = 0.157$) or entorhinal volume (indirect path: $z = -1.814$, $p = 0.070$).” (lines 296 – 312)

Discussion: “However, since synaptic dysfunction may not be the only mechanistic explanation behind our results, we additionally investigated the influence of other potential mechanisms including functional connectivity, WMH, and neural noise. We found that, even when including functional connectivity and WMH volume into a parallel mediation model in conjunction with measures of MTL volume, the association between encoding-related activation and cognition remained. Therefore, it appears unlikely that our disease-related activation was driven by white matter damage or functional connectivity. With regards to the investigation of noise or unexplained variability in the fMRI, we found that overlapping disease-related differences in (de-)activation and noise both explained cognitive variability, yet, the effect sizes for BOLD activity were much larger than for noise when incorporated into the same model. Specifically, we found that unexplained variability increased over AD disease stages. Those findings may be related to studies showing that AD is characterized by a reduction in neural selectivity⁵⁴, with potential mechanistic explanations being related to excitation-inhibition imbalance⁵⁵, dedifferentiation³⁴, or a breakdown of scaffolding³⁵ mechanisms.” (lines 422 – 435)

R1.03: Related to the point above, in Figure 4 the authors show that deactivation is most prominent at later disease stages i.e. increases for individuals above the median split of their continuous disease stage measure, and decreases for those below the median split for this measure. Though critical for their conclusions, this finding met the threshold for significance for deactivation but not activation, and was based on a median-split approach that was not justified. Can the authors explain why they dichotomized their continuous measure only for this analysis?

Response to R1.03: We thank the reviewer for raising this important point regarding the use of a median split in the subanalysis presented in Figure 4. We fully acknowledge that dichotomization of a continuous measure can reduce statistical power and inflate the probability of a Type I error (Altman & Royston, 2006), while equally relaxing the linearity assumption in the continuous analysis. In light of this concern, we have repeated the analysis using the continuous disease stages. The revised results closely replicate the findings obtained with the median split, thereby confirming the robustness of our conclusions while addressing the statistical limitations inherent in dichotomization. We updated the statistics according to the revised results from the continuous analysis. We also revised the method section.

Results: “If longitudinal data were reflective of our empirical disease curves for fMRI from above, this would provide further leverage for establishing fMRI as a biomarker in AD. To this end, we hypothesized that cross-sectional disease stage would moderate longitudinal changes in successful encoding-related activity across follow-ups, particularly in deactivation. For this analysis, we used longitudinal available fMRI data from four time points. Mean follow-up availability was different across the groups with more data available for CN than any other group ($F_{3,132.41} = 25.189$, $p < 0.001$, $\eta^2 = 0.10$ [0.06; 1]; Supplementary table 1). Random effect variances in the random-intercept-random-slope models were very close to zero, which is why we report on the random intercept models only. A potential reason may be the inherent problem of large intra-individual

variability known to occur in longitudinal task-fMRI^{45,46}. The corresponding tables for the random-intercept models can be found in the supplementary tables 8 and 9, respectively. As expected, we found change over follow-ups in encoding-related deactivation to be significantly moderated by disease stage ($t(346.969) = 2.716$, $p_{\text{uncorrected}} = .00693$, $p_{\text{Bonferroni-Holm}} = .01386$, $\eta^2_{\text{partial}} = .02$ [.00323;1]; Fig. 4A). On the other hand, this was not the case for encoding-related activations ($t(317.37) = 1.473$, $p_{\text{uncorrected}} = .141758$, $p_{\text{Bonferroni-Holm}} = .141758$, $\eta^2_{\text{partial}} = .006$ [0;1]; Fig. 4B)". (Lines 332 – 349)

Methods: "Lastly, we aimed at testing above findings longitudinally. First, we calculated linear mixed effect models (LMEs) in R with disease-related activation and deactivation as the separate dependent variables. The effect of interest was the interaction between time over follow-ups and continuous disease stage. Covariates were main effects (and interactions with time) of demographics age, sex and education. We estimated LMEs with and without time (slope) as random effect, but the former had singular fits, thus the latter models were used for the analysis." (lines 715 – 720)

R1.04: The disease progress model is an interesting approach but raises some questions that should be answered. What makes this approach Bayesian, is it the prior of a monotonic sigmoidal that the data must be fit along? How does this approach differ from other population trajectory modeling approaches like sampled iterative local approximation (SILA) and why was the DPM approach chosen for this study? Is it appropriate to include participants with just one time point in the DPM approach (as seen in Supplementary Figure 1) for fitting a longitudinal trajectory? Also, please include the model formula in the methods or supplement.

Response to R1.04: We thank the reviewer for their questions on the DPM approach chosen. In this answer, we will try to provide more details about the methodological background, what separates the DPM from other established approaches and why we believe that participants with a single time point merit inclusion in the fitting process. First, the Bayesian qualities of this particular DPM approach lie in the specific way the biomarker progression curves are modelled. Our motivation for this approach stems from previous publications with Bayesian and Gaussian Process (GP) approaches in the past (Ziegler et al., 2014, 2015, 2017, Lorenzi/Ziegler et al., 2015, Nemali et al., 2023). Gaussian Processes are fundamentally Bayesian in terms of describing the to be estimated regression curves in terms of priors over smooth functions (here even multivariate with numbers of biomarkers modelled) and their respective posteriors (after the data has been observed). GPs are very flexible non-parametric regression techniques (with trajectory complexity being a function of the dataset) and which have also the advantage of enabling probabilistic predictions (with analytical expressions), supporting uncertainty quantification in clinical settings (Rasmussen & Williams, 2006). The monotonicity is an additional constraint that was incorporated in the curve fitting process by Lorenzi et al. 2019, and the individual disease stage is a continuous latent variable (for which posterior expectations and variance are obtained), e.g. disease stage t of a participant j given the data. As such, both biomarker curves and disease stage come with expected means and uncertainties. To explain this in more detail, we have expanded the following paragraph in the method's section including model formulas.

Methods: "For quantification of each patient's disease stage during progression towards AD, we utilized a continuous multivariate DPM based on Bayesian Gaussian Process (GP) regression^{14,72} (code available from https://gitlab.inria.fr/epione/GP_progression_model_V2). In this approach introduced by Lorenzi et al.¹⁴, GPs were used to empirically model smooth monotonically increasing transition curves from normal to abnormal biomarker levels as proposed in Jack et al.⁴⁷ (see Lorenzi et al.¹⁴ for mathematical details). The model can be described as follows:

$$\mathbf{y}^j(t) = \left(y_{b_1}^j(t), y_{b_2}^j(t), \dots, y_{b_{N_b}}^j(t) \right)^T$$

$$= \mathbf{f}(t) + \mathbf{v}^j(t) + \epsilon,$$

where $y^j(t)$ are the longitudinal biomarker measurements at time point t , which can be represented by the fixed-effect model $f \sim GP(0, \Sigma_\epsilon)$, the individual random effects $\mathbf{v}^j(t) = (\mathbf{v}_{b_1}^j(t), \mathbf{v}_{b_2}^j(t), \dots, \mathbf{v}_{b_{N_b}}^j(t))^\top$ and $\epsilon = (\epsilon_{b_1}, \epsilon_{b_2}, \dots, \epsilon_{b_{N_b}})^\top$ observational residuals.

After time-reparameterization, the biomarker evolution over reparameterized disease time can be described as

$$\mathbf{y}^j(\phi^j(\tau)) = \mathbf{f}(\phi^j(\tau)) + \mathbf{v}^j(\phi^j(\tau)) + \epsilon,$$

where $\phi^j(\tau) = \tau + d^j$. In this model, d^j corresponds to the individual time-shift of a participant. For further mathematical details, please resort to Lorenzi et al.¹⁴. Our DPM application in this study focused on the following groups of variables based on the ATN pathological classification system and cognitive performance: (A) Biomarkers from CSF were A β 42/40 ratio and pTau₁₈₁ levels; (B) Atrophy measures comprised bilateral hippocampus and entorhinal cortex volumes from FreeSurfer (version 7, longitudinal pipeline, corrected for total intracranial volume (TICV)); and (C) Cognition was operationalized in terms of the PACC5 and the ADAS-COG-13 summary scores. Importantly, the model inversion based on observation of those features (including available follow-ups) estimates the most probable individual patient's disease stage continuously (including its uncertainty) and a data-driven disease progression curve for each of the fitted biomarkers." (Lines 633 – 659)

[...]

"In detail, according to Lorenzi et al.¹⁴, the estimated disease stage is given as the expected value of

$$p(t^* | y^*, y, t, m, t') = \frac{(p(y^* | t^*, y, t, m, t') * p(t^*))}{(p(y^* | y, t, m, t'))} \text{ (Lines 659 – 661)}$$

Response to R1.04 (continued): Second, sampled iterative local approximation (SILA) is a data driven model to estimate amyloid positivity onset in years using PET data (see Betthausen et al., 2022). We believe that one key difference lies in the multivariate input into the DPM utilized in this study vs. the univariate input to the SILA algorithm. The DPM we used by Lorenzi et al. (2019) provides a key advantage in that the resulting disease progression curves are generated in a multivariate manner inferring the progression in all input variables (i.e. multiple biomarker, volumetric, and cognitive scores) at the same time to estimate monotonic disease progression curves as well as individual disease stages. Modelling the transition and covariation across multiple biomarker domains better accounts for the nature of neurodegenerative diseases and likely improves estimates of the most probably individual stage. Rates of atrophy and biomarker accumulation are characterized by interindividual differences which partly depend on resilience factors such as brain maintenance or cognitive reserve. Based on all data, the DPM approach used here infers the most likely time point on the disease time frame including its uncertainty (Lorenzi et al., 2019). In SILA, time estimation is based on the population trend in change in amyloid accumulation over time. Time represents the estimated amyloid onset age (Betthausen et al., 2022), while disease time in the DPM used here represents a latent time variable given multivariate data (Lorenzi et al., 2019). Taken together with the above state experience with GPs and having collaborated with Dr. Lorenzi in the past on GPs, we focused on this particular approach to continuous disease modelling rather than using tools less unknown to us. We hope to perform empirical comparisons of various valuable approaches in the future with more project resources on DPMs per se.

We further appreciate the reviewer's concern regarding the inclusion of participants with a single observation in our disease progression modeling framework. However, we respectfully argue that their inclusion is both methodologically sound and practically valuable within the Gaussian Process (GP) modeling approach adopted here (Lorenzi et al., 2019). Unlike models that require explicit longitudinal follow-up for curve estimation, our GP-based framework models smooth, population-level biomarker trajectories over a latent disease time axis, which is inferred probabilistically across the cohort using both longitudinal and cross-sectional data. Even a single time point contributes meaningfully to this inference by constraining the GP posterior adaptively, particularly in less represented stages of disease. Moreover, in clinical datasets such as ADNI or the DELCODE sample utilized here, incomplete follow-up is common and excluding such data might not only reduce statistical power but could also slightly introduce bias by disproportionately underrepresenting early or advanced disease stages. We emphasize that the Bayesian formulation used here rigorously accounts for uncertainty and missing data (see Lorenzi et al., 2019), supports the inclusion of single-observation participants enhances—rather than compromises—the model's generalizability and robustness.

R1.05: Though the manuscript reads quite nicely, much important methods info is omitted that would be confusing for a reader going through the manuscript sequentially, without having to jump to the methods at the end. For example, associations with fMRI task performance are reported at the end of Section 2.1, but the task itself has not been described, nor how the A' measure was computed or what it indicates. It also not clear what exact data is being used for each analysis stage, e.g. how many fMRI time points are used to generate the activation trajectories in 2C-D. In some cases this information is in the figure legend but should be in the manuscript text, please check to make sure important methods info is not missing.

Response to R1.05: We appreciate and understand the reviewer's concern regarding the lack of methodological clarity early on in the manuscript. As a response, we have adjusted our manuscript by providing additional methodological insight in the introduction and results section of the manuscript. In detail, we have included a methods description of the fMRI task and the task performance measure at appropriate places. Further, we have added information on the numbers of participants and timepoints used to generate the results in the results text where space permitted. This is in line with an additional comment by reviewer

Introduction: *"We used task-fMRI data from a modified version of an incidental learning task²⁵, in which participants are instructed to classify scenes as indoor or outdoor scenes via a button press. Participants were pre-familiarized with one indoor and outdoor scene. During the task, participants were presented with those familiarized images as well as novel images. Following the fMRI task, participants completed a recognition task during which they had to rate their confidence about their classification of images as novel or familiar. Resulting from the novel and the confidence ratings towards the ratings is a fMRI marker for successful memory encoding which we used in all further analyses. More details can be found in the online methods."* (Lines 121 – 129)

Results: *"Lastly, we were interested in the association between disease stages and a fMRI task performance marker (A' – "A prime") not used in model fitting. Task performance was measured as the area under the curve of hits versus false alarms in the recognition task 70 min post fMRI task (see ³³ for details). While a score of .5 would indicate guessing randomly, larger scores reflect better performance. More details can be found in the method section."* (Lines 209 – 213)

Results: *"Since the progression of activity changes in patients transitioning towards AD exhibited unexpected dynamics (e.g. in early stages), we next assessed activity progression curves with a more flexible curve fitting approach using n = 493 data points (i.e. one value per participant)."* (Lines 240 – 242)

Online Methods: *"From this recognition test, the area under the receiver operating characteristic curve of hits (i.e. correctly classified novel images as novel) versus false alarms (i.e. falsely classified familiar images as novel) was calculated (A' – "A prime")³³: $A' = \int_0^1 H(FA)dFA$, with H and FA being defined as Hits or False Alarms, respectively."* (Lines 605 – 608)

Minor concerns:

R1.06: Tau positivity of the ATN framework was defined using ptau181, but this measure tends to be less predictive of tau pathology than ptau217, and tends to represent pathology change somewhere after amyloid accumulation but before significant tau pathology. Can the authors justify using this measure to define tau positivity?

Response to R1.06: We thank the reviewer for pointing this out. In fact, there is empirical evidence supporting that CSF ptau217 might perform better than ptau181 as a biomarker of tau pathology in context of AD classification (Janelidze et al., 2020; Leuzy et al., 2021). Unfortunately, in the DZNE DELCODE cohort, only the essays of ptau181 and total tau were available. Hence, we focused on ptau181 as a

biomarker for tau pathology. For clarity, we have added a short section within the limitations part of the discussion to make this point clear:

Limitations: “Third, we used CSF biomarkers instead of regional biomarker load (PET) resulting in a global estimate for amyloid and tau biomarker status. In particular, a previous study showed that local tau depositions are associated with local changes of activation in EM systems⁵⁶. Further, essays of pTau₂₁₇ were not available in the cohort analyzed. It has been recently shown that pTau₂₁₇ might outperform pTau₁₈₁ in terms of diagnostic performance^{57,58}.” (Lines 462 – 466)

R1.07: Was the sample used in the DPM analysis 208 individuals as stated in the text, or 210 as stated in the legend for Figure 1?

Response to R1.07: We thank the reviewer for this careful observation and apologize for this inconsistency. In the updated version of the manuscript, this error has been corrected. In total, 208 participants were used for the DPM fit. In the revised version of the manuscript we updated the legend of Fig. 1 accordingly as follows:

Figure Legend 1: “Disease progression curves and association with AD-related variables. A GP-based DPM used in this study comprising longitudinal CSF (A β 42/40, pTau₁₈₁), volumetric MRI (hippocampal, entorhinal volume), and PACC5 as well as ADAS-COG-13 cognitive score within the DELCODE cohort using 787 available data points from 208 participants (80 CN, 57 A+ SCD, 44 A+ MCI, 27 A+ DAT). B Timepoints of fastest changes derived from temporal derivatives of the empirical biomarker progression curves from the model posterior distribution. The timepoint of fastest change was sampling 200 times from the model posterior and the red line indicates its median. C Ridgeline plots of disease stages in diagnostic groups. D Ridgeline plots of disease stages in AT classification subgroups of AD pathology. E Associations between the estimated disease stage and memory performance during the task-fMRI session residualized for age, sex, and education. ***p < .001.”

R1.08: Why do the authors think sex was not related to more advanced disease stage, given the growing body of work suggesting that women accumulate AD pathology more quickly than men?

Response to R1.08: We thank the reviewer for the critical assessment of our results. We acknowledge the broad evidence for sex differences in AD research. Overall, among others, research has shown higher AD incidence for females (Chêne et al., 2014), lower functional connectivity (Williamson et al., 2024), greater overall tau burden (Barnes et al., 2005; Contador et al., 2021 for a study of early-onset AD; Buckley et al., 2019), and faster cognitive decline (for reviews, see Ferretti et al., 2018; Bonkhoff et al., 2025; Lopez-Lee et al., 2024). Additionally, faster accumulation rates were indeed reported for amyloid positive females in comparison to males (Smith et al., 2020; Wang et al., 2024). Our results showed that disease stages were not different between males and females. One explanation refers to the sex distributions in our sample. In the DELCODE sample, sexes are more or less evenly distributed between the different diagnostic groups (Jessen et al., 2018). Indeed, in our subsample of DELCODE participants, we only found that SCD participants included significantly lower relative amounts of females in comparison to normal controls (supplementary table 2). In the subsample used for model fit, sex distributions between diagnostic groups were not different as well (supplementary table 1). Thus, after model fit, it is not unlikely that sex differences were not found because of a sampling artefact. If sex distributions in the overall sample and diagnostic groups are not different, similar sex distributions are likely to show up for the disease stage variable. Furthermore, sex differences can present with local differences in biomarker accumulation across the brain (Wang et al., 2024), which were not modelled using the global estimates of biomarker burden of CSF data. Further, our DPM integrates longitudinal biomarker, structural MRI, and cognitive data to improve disease stage estimation. This is a strength of our approach, as it allows for more precise staging based on observed trajectories over time rather than single time-point snapshots while also adding multivariate information from multiple sources of variation across the AD spectrum. However, while the model incorporates longitudinal data to estimate the expected disease stage, it assumes a single, shared progression trajectory across participants (Lorenzi et al., 2019). This assumption implicitly models all individuals as progressing through the disease at the same pace. As such, individual or subgroup-level

variability — including sex-specific differences in progression speed or order of biomarker changes — may be absorbed into model uncertainty or residuals rather than expressed as systematic differences in stage. Moreover, the disease stage output of the DPM reflects an individual's location along a canonical disease timeline, but it may not capture divergent progression pathways or nonlinearities that could differ by sex. For example, as referenced above, some studies suggest that women may exhibit faster tau accumulation or steeper cognitive decline in later stages — dynamics that might not shift overall disease stage unless explicitly modeled. In this study, we decided against a subgroup analysis, because we were interested in global trends in functional differences across the full clinical AD risk spectrum. Therefore, we have added a paragraph in the discussion section to reference this point.

Limitations: *“Of particular note, we did not observe sex differences in our study in terms of disease stages, even though a large body of research suggests differential cognitive decline^{53,54}, biomarker accumulation^{55,56}, and incidence rates for females compared to males⁵⁷. This may be related to a sampling effect as both, our fitting sample (supplementary table 1) and overall analysis sample (supplementary table 2) only presented with minor differences in sex distributions across diagnostic groups. Further, based on the DPM conceptualization, as all participants are hypothesized to progress through the disease with the same speed⁵⁴. Thus, the disease stages are informative about the relative position on the latent disease time axis but do not inform about differential speed of decline.” (Lines 466 – 475)*

R1.09: Do the density plots in Figure 1C and 1D only show baseline data?

Response to R1.09: *We thank the reviewer for pointing this out. In fact, the density plots in Figures 1C and 1D show baseline data, indeed. We have added a respective statement to the legend of figure 1:*

Figure 1 Legend: *“[...] C Ridgeline plots of cross-sectional disease stages in diagnostic groups of the full analysis sample (n = 493). D Ridgeline plots of cross-sectional disease stages in AT classification subgroups of AD pathology (n = 222) [...]”*

R1.10: What is the p-value threshold for the voxelwise results displayed in 2B (not stated in text, hard to tell from legend)?

Response to R1.10: We have added the respective threshold in the results section as well in the accompanying legend of figure 2.

Results section 2.2: *“We observed that more advanced disease scores were related to hyperactivation in the precuneus, inferior parietal lobule, and posterior cingulate cortices bilaterally, as well as anterior cingulate cortex and superior frontal gyrus (Fig. 2B, orange colors, Family-Wise error (FWE) corrected with $p_{FWE} < 0.05$), reflecting similar previous observations based on the novelty contrast of the same paradigm²⁷. Additionally, we found right-lateralized reductions in encoding-related activation with advancing disease progression within the middle occipital gyrus and inferior temporal gyrus (Fig. 2B, blue colors, $p_{FWE} < 0.05$).” (Lines 226 – 233)*

Legend figure 2: *“[...] B Linear associations of disease stage with successful memory encoding related activation (red colors) and deactivation (blue colors) from the second-level SwE model. Results were obtained using the Wild bootstrap method with 1000 repetitions thresholded at $p_{FWE} < 0.05$. [...]”*

R1.11: Are the results described in section 2.4 adjusted for age, sex, and education? If not, could most of these results be explained by an association with age?

Response to R1.11: *Yes, the results are corrected for demographics age, sex, and years of education. To make this clear, we have added the following passages to the full text:*

Results: “The disease stage is a marginal score combining the influence of several components into one single index. Thus, we were further interested in individual contributions of biomarkers towards activity variability beyond demographic information (age, sex, and years of education). [...]” (Lines 258 – 260)

R1.12: For clarity, figure 4B should show the same trajectories as 4A i.e. with the early and late stage groups separated, even if the result is not significant.

Response to R1.12: We thank the reviewer for this recommendation. In response, we have included the non-significant group-specific longitudinal slope estimates. We have additionally removed the sentence specifying why we report on the overall slope over time. Additionally, in accordance with suggestions made by the reviewer in R1.03, we made changes to the longitudinal analysis by including continuous disease stage values in the analysis.

Figure 4:

Figure 4. Association between disease stage and longitudinal change of successful memory encoding-related activation and deactivation. For this analysis, longitudinal fMRI data from $n = 166$ individuals with available AD biomarker information was used ($n_{A.T.} = 108, n_{A+.T.} = 33, n_{A+.T.} = 27$). For visualization of the interaction, participants were grouped into early and late stages based on the median disease stage (-3.37) in the sample, such that each participant with a disease stage smaller than -3.37 was defined “in early stages” whereas every participant with a higher disease stage value was defined as “in late stages” of AD. **A** Interaction effect Time x Disease Stage on disease-related encoding deactivation. **B** The same interaction effect as tested in A, but for disease-related encoding activation. Here, change over follow-ups is not significantly moderated by disease stage. P-values were corrected using the Bonferroni-Holm procedure. * $p_{Holm} < .05$; $n.s. p_{Holm} > .05$.

R1.13: In view of less pronounced activity differences between successful encoding with declining cognition, the authors should mention the literature around network dedifferentiation in aging and AD in the discussion section.

Response to R1.13: We thank the reviewer for bringing the dedifferentiation literature to our attention. Indeed, the dedifferentiation hypothesis postulates a breakdown of sophisticated distinction in the responses towards stimuli. However, our evidence regarding dedifferentiation is incomplete. While we did find diminishing activity differences for successfully remembered stimuli in areas of deactivation and activation, respectively, we did not find other regions showing a subsequent memory effect. Nevertheless, we added a section in the discussion referring to the dedifferentiation as one of the potential mechanisms behind our activation disease stage association.

Discussion: “Additionally, our results may be in line with the dedifferentiation hypothesis of cognitive aging^{34,49}. According to the dedifferentiation hypothesis of cognitive aging, brain areas might fail to respond in a selective and specific manner towards different stimuli. Since the underlying successful memory contrast in our study is an indication of differential activation towards successfully encoded vs. non-encoded items, it may be plausible that the EM network experiences a reduction in specificity to successfully encoded vs. non-encoded stimuli, giving rise to cognitive decline via dedifferentiation. Similar results were reported earlier in a study investigating functional connectivity in preclinical AD⁵⁰. However, we did not find other regions outside the episodic memory network suddenly showing a subsequent memory effect along the disease cascade. Thus, the evidence for dedifferentiation in our analysis remains incomplete.” (Lines 388 – 398)

R1.14: In the discussion section Fig. 2C is said to show fMRI deactivations becoming abnormal after CSF and MTL volume change, but the orange line appears to increase prior to any of these other variables?

Response to R1.14: We thank the reviewer for the careful observation of our results. Indeed, in the beginning of the cascade, our results suggest initial changes in deactivation abnormality. Further, we would like the reviewer to draw their attention to our response to R1.01, in which we also discussed this observation in more depth. However, for comparability, we used timepoints of fastest change (i.e. the time point a function gradient is maximal) as a mathematical landmark. This facilitates the interpretation about where the maximum change along the disease trajectory occurs. In order to provide more detail on this, we added a supplementary figure showing first derivatives of the disease progression curves. The maximum of each curve represents the point estimates we reported in the main text.

Reviewer #2

This study presents an ambitious and well-structured investigation into episodic memory network dysfunction in the aging - Alzheimer's disease (AD) continuum, using an extensive and enviable dataset from the DELCODE study. The authors integrate task-based fMRI – as a proxy of synaptic dysfunction - within the amyloid-cascade disease progression model (DPM). The authors showed widespread loss of deactivation and activation with disease progression which related with cognitive decline (partially independent of neurodegeneration – as quantified by regional atrophy). The authors conclude that synaptic dysfunction and neurodegeneration are independent drivers of cognitive decline. The methods are generally state-of-the-art, and the conceptual framework is generally clear. I have however several conceptual and methodological concerns that I think require further clarification.

Major comments

R2.01: Operational circularity in defining synaptic dysfunction. The study defines synaptic dysfunction in terms of the fMRI signal changes that are being measured. This creates a risk of conflating the measurement (i.e., fMRI signal) with the underlying physiological construct of synaptic dysfunction. Interindividual fMRI variability in the context of aging and disease can reflect multiple factors—such as individual cognitive strategies, neurodegeneration (which is partly controlled for), vascular changes, and other noise sources. The assumption that fMRI-activity reflects synaptic dysfunction is central to the manuscript's claims and appears either questionable or circular. The interpretation of the findings may therefore be a bit forced and oversimplistic.

Response to R2.01: We thank the reviewer for raising this critical point. Indeed, we agree with the reviewer's opinion that the initial version of the introductory section and discussion seemed forced and oversimplistic. In line with this respective comment and the following comment by the same reviewer, we rewrote parts of the introduction and discussion in order to take explanatory factors other than synaptic dysfunction into account.

Introduction: "Hyper- and hypoactivation-like patterns have been observed in human task-fMRI studies of AD. These studies have reported lower deactivations in regions normally deactivated (also denoted as "hyperactivation" and "reduced deactivation", respectively) during memory tasks like novelty detection or successful encoding, most notably structures of the DMN like the posterior cingulate cortex and precuneus^{24,26,27}. For areas activated rather than deactivated during memory tasks (e.g., the inferior temporal lobe and hippocampus), a number of studies have reported hypoactivation^{8,28,29}. Explanations for this aberrant activity stem from human and animal research. For example, studies using AD mouse models have shown hyperactivation of neurons in direct vicinity of soluble and insoluble A β deposits compared to neurons distant from deposits in the hippocampus^{30,31}. In the face of combined A β - and tau pathology, on the other hand, neurons show hypoactivation⁵. One mechanism underlying this activation differences may be synaptic dysfunction. In animal research, synaptic dysfunction has been conceptualized as network dysfunction independent from synapse loss⁵. In line with this rationale, a recent fMRI study has found brain activity abnormality that cannot be explained by neurodegeneration⁸.

However, while synaptic dysfunction may be one potential candidate as explanatory factor of brain activity abnormalities in the human fMRI, further mechanisms have been proposed. For example, studies have provided evidence that vascular pathology³² or functional connectivity³³ can account for BOLD signal changes. Furthermore, in the cognitive neuroscience literature, dedifferentiation³⁴ and scaffolding³⁵ mechanisms are discussed. There, it is still not clear in how far activation abnormalities present as compensatory^{35,36} or detrimental. Here, we use atrophy, vascular pathology and functional connectivity in conjunction with activity from EM to investigate EM network abnormalities in human task-based fMRI in AD. Our study builds on reported brain activation abnormalities to model their relationship to AD progression with a DPM." (Lines 130 – 152)

Discussion: "However, since synaptic dysfunction may not be the only mechanistic explanation behind our results, we additionally investigated the influence of other potential mechanisms including functional connectivity, WMH, and neural noise. We found that, even when including functional connectivity and WMH volume into a parallel mediation model in conjunction with measures of MTL volume, the association between encoding-related activation and cognition remained. Therefore, it appears unlikely that our disease-related activation was driven by white matter damage or functional connectivity. With regards to the investigation of noise or unexplained variability in the fMRI, we found that overlapping disease-related differences in (de-)activation and noise both explained cognitive variability, yet, the effect sizes for BOLD activity were much larger than for noise when incorporated into the same model. Specifically, we found that unexplained variability increased over AD disease stages. Those findings may be related to studies showing that AD is characterized by a reduction in neural selectivity⁵⁴, with potential mechanistic explanations being related to excitation-inhibition imbalance⁵⁵, dedifferentiation³⁴, or a breakdown of scaffolding³⁵ mechanisms." (Lines 428 – 441)

We agree with the reviewer that interindividual fMRI variability in the context of aging and disease can reflect multiple factors. For example, hippocampal atrophy in the CA3 region has been shown to be related to hippocampal hyperactivation in a memory task (Yassa et al., 2010). Further, recent advances have provided insight into the effect of cerebrovascular reactivity on the BOLD response (e.g. Tang et al., 2025). We further thank the reviewer for bringing this important point to our attention and regret we did not provide information on this matter in the initially submitted manuscript. During the revision, we aimed at providing an estimate of alternative explanations for our association between activity and disease progression scores by taking volumetry, vascular pathology and functional connectivity into account all at once. This may help to uncover potential explanations of our effects which go beyond the assumption of synaptic dysfunction. To conform with this aim, we took volumetry, vascular and functional connectivity measures into account by running parallel mediation analyses. More specifically, we used hippocampal and entorhinal volume, as well as total white matter hyperintensity volumes and effective connectivity as mediators for the association between fMRI-based episodic memory activity and memory outcomes. Even though both volumes, WMH and effective connectivity were included in the model, the direct effect between activation and deactivation and

cognition remained statistically significant, respectively. Model fit was excellent. We used 407 participants from our baseline data set with complete volume, activation, cognition, WMH, and connectivity info. Those results are detailed in the revised manuscript parts found below. While an integration of those results provides reasons to include synaptic dysfunction as a potential explanation of our association between disease stage and deactivation, we cannot neglect different mechanistic explanations that were not targeted in the initial version of the manuscript.

Supplementary Figure 4:

With regards to the cognitive strategies topic, we regret that any behavioral data or questionnaires pertaining to cognitive strategies were not recorded. However, we estimated putative cognitive strategy variability. For this, we first regressed out fMRI performance and demographics age, sex, and years of education of the contrast images using multiple regression in SPM. The residual participant contrast images were then submitted to a principal component analysis for data reduction. Since the contrast images may still contain unwanted variability, we inspected the components and discarded any which were classified as artefactual. Next, the remaining PCs were submitted to a Gaussian Mixture Modelling algorithm to identify any clusters in the data. We did not find any clustering in our data based on putative cognitive strategies.

Results: “In order to discern whether the associations survive the correction for additional third variables, we performed parallel mediation analyses including volume, white matter hyperintensities (WMH), and effective connectivity. Effective connectivity values were obtained from a recent study utilizing dynamic causal modeling⁴⁴. WMH were log-transformed and all variables entering the model were corrected for baseline age, sex, and years of education a priori. Supplementary Figure 5 provides an illustration of this mediation analysis. The complete model outputs including fit indices can be found in the supplementary tables 5 and 6. For disease-related activation, hippocampal volume (indirect path: $z = 2.102$, $p = 0.036$) and effective connectivity (indirect path: $z = 2.057$, $p = 0.040$) mediated the association between activation and cognition. Entorhinal volume (indirect path: $z = 1.580$, $p = 0.114$) and WMH (indirect path: $z = 1.023$, $p = 0.306$), on the other hand, did not significantly mediate the association. Even if mediators were taken into account, higher levels of cognition were related to higher activation (direct path: $z = 4.117$, $p < 0.001$). For disease-related deactivation (Supplementary Figure 4B), again, hippocampal volume (indirect path: $z = -2.211$, $p = 0.027$) and effective connectivity (indirect path: $z = -2.354$, $p = 0.019$) partially mediated the association between deactivation and cognition (direct path: $z = -4.799$; $p < 0.001$), but not WMH (indirect path: $z = -1.414$, $p = 0.157$) or entorhinal volume (indirect path: $z = -1.814$, $p = 0.070$).

Finally, we also tested for whether differences in residual variability could explain our fMRI results. Instead of the contrast images as in Figure 2B, we submitted the residual variability maps from the first level fMRI modeling to the second-level SwE utilizing the same design matrix as above. We found that there was a wide spread increase in residual variability over the disease stages after controlling for FWE (threshold $p < 0.05$; Figure 4A). The disease-related differences also partly overlapped with the disease-related activation results from section 2.2. To determine whether noise or the activity values predominates in the disease stage association, we extracted baseline contrast values and noise estimates from the overlapping regions and submitted them to a multiple regression analysis controlling for age, sex, and years of education with A' as the dependent variable. Result tables can be found the supplementary tables 7 and 8.” (lines 297 – 323)

Online Methods: “White matter hyperintensities and effective connectivity

In our analysis of potential determinants of our results, we also included total white matter hyperintensity volumes using estimates from a previous study⁷⁷. WMH Volumes were segmented using the AI-augmented version of the Lesion Segmentation Toolbox and both T1-weighted MPRAGE and T2-weighted FLAIR images.

For our effective connectivity metric, we used the effective connectivity estimate from the right parahippocampal place area (PPA) to the right precuneus (PCU) from a previous DELCODE study using dynamic causal modeling (DCM)⁴⁴. This study modelled our task-fMRI data and investigated influences of CSF biomarkers on task-based effective connectivity of a-priori specified regions of interest. Extracted time-series data were corrected for age, sex, and years of education. Results showed that the effective connectivity from the right PPA onto the right PCU was modulated by the interaction between CSF biomarkers for amyloid and tau. Further details on the modeling approach and results can be found in Suksangharn et al.⁴⁴ (lines 676 – 688)

”

R2.02: Choice of Parametric Modulator for Signal Variability. The authors employ a parametric modulator of successful encoding as a proxy for synaptic dysfunction. It is unclear why this specific measure was selected over other potential fMRI metrics. Since AD participants tend to successfully encode fewer items, regression towards the mean could partially account for the observed results. Moreover, increased noise in AD individuals might also explain the relationship between task-fMRI metrics and the AD continuum.

Response to R2.02: We thank the reviewer for bringing up this important issue. In the conducted brain activation study using the FADE-paradigm in DELCODE participants, the successful memory encoding contrast can be calculated in several separate ways. In one conventional (so called categorical) method, only the trials can be contrasted differentiating activation differences between subsequently remembered vs. forgotten trials. However, previous studies have suggested that the degree to which items are later remembered might be related to the actual degree of activation of the episodic memory network in some functional sense (Soch et al., 2021, 2024, Fehlmann et al., 2020, Kizilirmak et al., 2024). Since the question can be seen as being about the specific structure of a first level task-fMRI model, it can be reformulated as an empirical question of model selection. Interestingly, it was shown in a recent study using Bayesian model selection that the parametric modulation is preferred over the categorical representation of the contrast (Soch et al., 2021). That study suggests that variability in old age fMRI activations during memory encoding might be more efficiently encoded using such a parametric modulator in first level GLMs (also resulting in less unexplained signal variability). Based on these quantitative and evidence-based recommendations we therefore opted for the parametric modulator while being aware of the alternatives. To make this clear, we have included the following statement in the methods section:

Online Methods: “*The choice for the parametric modulator was motivated by a recent study performing a model comparison identifying the parametric modulator as an improved characterization of encoding-related activations in old age in comparison to the standard categorical alternative of this contrast⁶⁸. ” (Lines 627 – 631)*

Moreover, the reviewer is concerned with potential regression towards the mean in the contrast modulated via the respective task performance. We thank the reviewer for the careful consideration of this confounding factor. We agree with the reviewer that the behavioral performance used for the parametric modulation shows interindividual variability. While the average response may be different, the variability of responses may be subject to considerable interindividual as well. If that were true, for example via a response bias due to fewer items being encoded, then the parametric modulation may be compromised. In order to address the reviewer’s concern, we have opted for three strategies. For one, we repeated the second-level GLM with the exclusion of AD participants (Supplementary Figure B). Thereby, we intended to investigate the potential influence of response biases shown by the AD group. If the behavioral responses were to account

for the second-level results, a different distribution of significant voxels would be expected than in our initial results using the parametric modulator (Supplementary figure A). We found that, while the effects for deactivation were largely preserved in this subsample, the results for activation were rendered non-significant. Even though this might indicate a potential regression towards the mean as suggested by the reviewer, we would like to respectfully argue that this may be related to a loss of power because of reduced sample size (due to the exclusion of AD participants).

Second, to further investigate robustness of our results in the initial version of the manuscript, we performed another second-level GLM using the Wild Bootstrap method and family-wise error correction. However, this time, we used contrast images from a conventional categorical approach of first-level fMRI modeling. Here, the effects of interest were not a collapsed novelty regressor, a parametric modulator, and a regressor for the master images as in the initial version of the manuscript, but rather three regressors being concerned with novel images being forgotten, remembered, and a master images regressor. The categorization of remembered vs forgotten was achieved by grouping novel stimuli based on the responses in the fMRI task. The contrast used for the second-level therefore was the difference between later remembered and later forgotten novel stimuli. We found that, even though we used a different modeling approach, the results were largely consistent with the parametric results from the initial version of the manuscript (Supplementary Figure C). However, we also note that in this second-level GLM with the categorical first-level contrast images, the activation results are not present.

In a third analysis step, we investigated behavioral performance variability directly. We reasoned that, if behavioral performance has an effect on our previous results, then the inclusion of a between-subjects variable targeting response variability into the second-level GLM would potentially render our initial results non-significant. For this, we extracted the behavioral variability by calculating standard deviations across all behavioral responses from the fMRI task. Those were then entered as a covariate into the second-level GLM. When investigating the association between activation and disease stages while controlling for response variability, our initial results were largely confirmed (see Supplementary Figure A and D). In summary, while we would argue in favor of robustness in relation to the deactivation results, the results on the activation areas are more heterogenous. Thus, we added the following sections in the results and discussion of the manuscript:

Discussion: “Fifth, as the activation differences along the disease progression vanished when excluding AD participants from the model and since this subsample is prone to significantly encode fewer items, we cannot rule out that the reduction in activation may stem from a regression towards the mean. On the other hand, a loss of power due to an exclusion of participants may also explain the null findings for activation when excluding AD participants. Additionally, we aimed at mitigating this already by excluding participants with a response bias prior to entering them in the analysis.” (Lines 484 – 490)

Finally, the reviewer is concerned that increased noise in the AD participants may explain our results. We thank the reviewer for bringing up this important point. Indeed, cognitive performance is deteriorated in AD with more false-alarms in comparison to healthy older adults (Düzel et al., 2011; Budson et al., 2006; Hildebrandt et al., 2009). This may prove to be a challenge for the estimation of a parametrically modulated design and may ultimately lead to larger unexplained variance in AD participants in comparison to controls. To answer the question of a putative confounding effect of noise, we have investigated the residual variance maps from SPM using the full sample to answer this question. For this, we calculated the exact same Wild Bootstrapped second level GLM as a function disease stage as in the main analysis of this paper. This time, however, the dependent scans were the ResMS.nii images. We found that there were considerable differences of unexplained variability across disease stages indicating increased BOLD signal noise levels with advanced disease progression (see supplementary figure E & F). It can be seen that regions across the brain show some overlap (Supplementary figure F) between the effects of noise (supplementary Figure E) and the activation differences across disease progression (Supplementary Figure A). Of particular interest are the red regions in supplementary figure F indicating the overlap between figure 2C from the paper and the noise effect. The overlap was calculated by means of the cosine angle between the two vectorized mask images. We found that $\cos(\theta) = 0.24$. Thus, while there is overlap between the two significant regions across disease progression, there are many regions that either only show the fMRI effect or the noise effect. Further, to answer the question whether noise confounds the results in the paper, we analyzed the overlapping regions and their activation and noise values by submitting both to a multiple linear regression predicting disease progression scores controlled for age, sex, and education. We found that the effect for activation ($t_{\text{activation}} = -3.907$, $p = .00108$; $\eta^2_{\text{partial}} = .0305$) was stronger than for noise in activation regions ($t_{\text{noise}} = 2.768$, $p = .0059$; $\eta^2_{\text{partial}} = .0177$). The same was true for deactivation regions ($t_{\text{deactivation}} = 4.900$, $p < .0001$, $\eta^2_{\text{partial}} = .0518$; $t_{\text{noise}} = 2.647$, $p = .0089$, $\eta^2_{\text{partial}} = .0221$). Additionally, both interaction terms between noise in the regions and the underlying activation were not significant. Because the parameters for the interaction were not zero, we concluded that a noise increase over the disease progress does not fully explain the activation and deactivation losses we observed in our study. Ultimately, with the follow-up analyzes presented above, we hoped to gain insight into potential concepts that may have compromised our initial results. Conclusively, results presented in the initial manuscript were largely preserved when investigating influences of response variability or modeling choice of the first-level GLM. Further, we did not find support for compromising effects of noise in a direct comparison of disease-related activation and disease-related noise increases. Since this additional analysis was a major part of the revision, we decided to include a dedicated results section into the manuscript. Additionally, we extended our discussion towards the raised points of response variability, modelling choice and noise.

R2.02.1: Relatedly (and very minor comment; out of curiosity), activation and deactivation measures have been combined in the past. Why are now reported separately? Is there any theoretical ground; if so, please explain.

Response to 2.02.1: We analyzed task-related activations and deactivations separately because they often reflect functionally distinct neural processes. Activations often involve task-relevant networks engaged by external stimuli or cognitive demands, while deactivations are frequently associated with the default mode network (DMN), reflecting a shift away from internally directed processes (Raichle et al., 2001; Shulman et al., 1997; Buckner et al., 2008). Prior work has demonstrated that task-related activations and deactivations can show opposite patterns of task modulation, and their independent analysis can reveal insights into cognitive control, attention, and clinical variability (Fox et al., 2005; Kelly et al., 2008; Anticevic et al., 2012). Analyzing both separately may help finding suitable interpretations of BOLD signal changes and might avoid averaging out meaningful differences of directions of effects (Spreng, 2012).

As a response, we have added the following section to the **Online Methods**: “*Task-related activations and deactivations were analyzed separately, as they reflect functionally distinct processes: activations typically involve task-relevant networks, whereas deactivations are often associated with the default mode network and disengagement from internally directed cognition*^{19,77}. Treating them separately avoids averaging out potentially opposing effects and enables a more precise interpretation of BOLD signal changes^{78–80}” (Lines 701 – 705)

Minor comments

R2.03: Clarity in Presenting Data and Task Description. The presentation of the results is somewhat difficult to follow because the details of the data and the specifics of the fMRI task are not introduced early in the manuscript. Providing a more thorough early description of the experimental design and the fMRI task would improve the reader’s understanding. Additionally, effect sizes are inconsistently reported; including them consistently would enhance the interpretability of the results.

Response to R2.03: We thank the reviewer for their comment and apologize for the lack of clarity of our task design early on in the manuscript. We agree with the reviewer that an earlier presentation of the task may aid understanding of our results presented further in the manuscript. Therefore, we have incorporated a description of the fMRI task in the introduction of the manuscript. We thereby hope to enhance the clarity about the fMRI results presented in the revised manuscript.

Introduction: “*We used task-fMRI data from a modified version of an incidental learning task²⁵, in which participants are instructed to classify scenes as indoor or outdoor scenes via a button press. Participants were pre-familiarized with one indoor and outdoor scene. During the task, participants were presented with those familiarized images as well as novel images. Following the fMRI task, participants completed a recognition task during which they had to rate their confidence about their classification of images as novel or familiar. Resulting from the novel and the confidence ratings towards the ratings is a fMRI marker for successful memory encoding which we used in all further analyses. More details can be found in the online methods.*” (Lines 121 – 129)

We additionally thank the reviewer for bringing the missing effect sizes at respective places to our attention. In the updated version of the manuscript, we now include effect sizes for all analyses.

Results: “[...] Investigating the associations of our disease stage scores with demographics in a multiple regression analysis, we found demographics to contribute significantly to the disease stage variability ($F(3,489) = 42.56, p < .001, \text{adjusted } R^2 = 0.202$). Specifically, higher age was indicative of a more advanced disease stage ($t = 9.148, p < .001, \eta^2_{\text{partial}} = .17 [0.13, 1]$), whereas higher education was related to earlier disease stages ($t = -4.979, p < .001, \eta^2_{\text{partial}} = .05 [0.02, 1]$). Sex showed no significant association with estimated disease stage ($t = 0.637, p = .524, \eta^2_{\text{partial}} = .0005 [0, 1]$).” (Lines 192 – 198)

“[...] Diagnostic group membership was significantly related to disease stage ($F(3,479) = 133.11, p < .001, \eta^2_{\text{partial}} = .46 [0.4, 1]$), reflecting the a priori assignment of participants to diagnostic groups at baseline.” (Line 202)

“[...] Additionally, when analyzing disease stage differences over the AT criteria, biomarker groups were significantly associated with disease stage ($F(2,209) = 92.11, p < .001, \eta^2_{\text{partial}} = .47 [0.39, 1]$).” (Line 206)

“ [...] Deactivation followed a non-linear progression trajectory, which was characterized by an initial increase of abnormality in the earliest disease stages, followed by the most pronounced changes in later disease stages ($F_{\text{non-linear}}(1.739, 489.216) = 4.567, p_{\text{uncorr}} = .01149, p_{\text{Holm}} = .02298, \eta^2_{\text{partial}} = 0.016 [0, 1]$; Fig. 2C).” (Lines 246 – 247)

“[...] Disease-related changes in fMRI activations resembled a monotonic increase as well, yet the non-linear effects were not significant ($F_{\text{non-linear}}(1.368, 489.632) = 2.931, p_{\text{uncorr}} = .07389, p_{\text{Holm}} = .07389, \eta^2_{\text{partial}} = 0.00812 [0, 1]$; Fig. 2D).” (Lines 252 – 254)

"[...] As expected, we found change over follow-ups in encoding-related deactivation to be significantly moderated by disease stage ($t(346.969) = 2.716$, $p_{uncorrected} = .00693$, $p_{Bonferroni-Holm} = .01386$, $\eta^2_{partial} = .02$ [.00323;1]; Fig. 4A). On the other hand, this was not the case for encoding-related activations ($t(317.37) = 1.473$, $p_{uncorrected} = .141758$, $p_{Bonferroni-Holm} = .141758$, $\eta^2_{partial} = .006$ [0;1]; Fig. 4B)." (Lines 346 – 349)

R2.04: While the DPM model uses longitudinal data; the fitting is highly dependent on cross-sectional information. I think this probably needs to be acknowledged. This might be harsh, as the DPM models have mostly relied on cross-sectional data. yet the results are highly dependent on the relationship between "cross-sectional" and "longitudinal" distributions + reliability of the measure. Think of memory or hippocampal volume in towards the healthy spectrum: the large variability in cross-sectional data can obscure real changes. The small variability cross-sectional in some biomarkers at baseline levels imply that small changes will be detectable quite early.

Response to R2.04: We thank the reviewer for this important comment. We agree with the reviewer that the fitting process is dependent on both cross-sectional and longitudinal information. Further, we also agree that different biomarkers show different ranges of baseline heterogeneity when all considered in the same space (see supplementary figure 2 - AD biomarkers varying less than cognitive performance markers when viewed on the same scale). This might be challenging for the data fitting process and result in stronger reliance on low-variability features. Additionally, previous DPM studies have also pointed out that in vivo modeling of biomarkers is ultimately contingent on the measurement noise of each biomarker (Oxtoby et al., 2018). Overreliance on cross-sectional effects where individual differences and noise are confounded might render progression estimates biased towards less variable and more reliable features.

We have included a paragraph in the limitation section in the discussion as follows:

Limitations: *"Additionally, model fit in DPMs is dependent on the measurement noise of utilized biomarkers. A previous study showed that measurement noise in CSF biomarkers can obscure group-level trends such that biomarker trajectories could not be fitted¹². Furthermore, large cross-sectional variability can obscure longitudinal trends which may result in a greater the emphasis of cross-sectional in comparison to longitudinal information in the fitting process of the DPM." (lines 460 – 465)*

R2.05: The DPM model and the later correlations with disease-ages are performed in partially overlapping samples. Please, clarify when and if there is some circularity in the analyses. I might have missed it; but are the activation and deactivation regions constrained by the Disease Stage Association model?

Response to R2.05: We thank the reviewer for this comment. We agree with the reviewer that subjects used in the model fit were also included in following analyses. This is true for roughly 50% of the model fitting subjects, since they additionally had available quality-controlled fMRI data. Again, we would like to reiterate that fMRI data was not used during model fitting. However, we would also like to note that the subjects used in model fitting likely have better disease stage estimates than subjects not used in model fitting, due to the amount of available DPM marker information. One reason for this is that model fitting subjects have at least one complete measurement occasion of all biomarkers, which is not true for the remaining subjects. Therefore, the disease stages of subjects in the model fitting sample are on the one hand more precise because of the amount of data used to generate the estimates (see Lorenzi et al., 2019). Nevertheless, to check for potential inflated statistics due to the inclusion of model fitting subjects, we repeated our voxel-wise association in the subset of participants with activation data not belonging to the fitting subset. This was done because the resulting regions extracted from this second-level GLM were used in consecutive analyses. As shown below, the significant regions are now constrained to the right supramarginal gyrus, only.

Revision Figure Reviewer 3. The effect of excluding model fit subjects from the analysis. In this figure, we discuss the exclusion of model fit participants from further analyses. The full sample contained 493 participants. Excluding the model fit participants who had available fMRI data left a sample of 349 individuals ($N_{CN} = 100, N_{SCD} = 177, N_{MCI} = 54, N_{AD} = 18$). **A** Comparison of effects from the initial manuscript of disease stage on brain activation using the full sample (left) and restricting the participants to the ones not used in model fitting. Here, results are constrained to the right supramarginal gyrus which is the same region of strongest effects as in the initial results. **B** Correlations between disease stage and deactivation values for the clusters in red in A. Excluding the model fit subjects (blue line) did not significantly change the correlation. **C** Similar results for the activation clusters (blue areas in A). When comparing the correlations between the full sample (red line) and the subjects not used in model fitting (blue line), the correlations are not significantly different from each other. Therefore, we conclude that an inclusion of model fitting subjects does not lead to a dramatic inflation of effect size.

All other effects were rendered non-significant. This is, however, also likely due to the exclusion of fMRI 300 scans stemming from the model fitting sample and thus, may also be related to a significant loss of power since the remaining significant voxels represent the voxels with the strongest effects in the full sample analysis as well. Additionally, and in response to the reviewer's comment regarding the constrained activation and deactivation regions, we note that the activation and deactivation regions are constrained by the GLM that associates memory encoding BOLD contrast to disease stage scores. Thus, we first fitted a DPM containing AD biomarkers, MTL volumes and cognition, only, and then used the resulting disease stage scores from this characterization of AD progression to predict activity differences in task-fMRI in the whole sample. We opted for this because of the significant noise in the fMRI contrast values prevented direct inclusion of fMRI in the DPM. Thus, while some subjects from the model fitting sample also had fMRI data, we did not use the fMRI data itself in conceptualization of the DPM. We regret that this was not clearly conveyed in the initial version of the manuscript. Therefore, we have added the following statement in the revised version of the manuscript in *Online Methods*:

Online Methods: "After establishing a score of disease progression for AD, we were interested in the association of that progression score with fMRI BOLD signal contrasts of memory encoding on the voxel-

level. We expected a region-specific association of activity with disease progression. The analysis was carried out using the Sandwich estimator toolbox⁷⁵ (v2.2.2, MATLAB). All available longitudinal scans were included in the fitting process to obtain between-subjects effects and additionally estimate longitudinal effects of time.” (lines 704 – 709)

Reviewer #3:

This study examines the relationship between brain function and markers of Alzheimer’s disease in 493 participants from the DELCODE study. Using CSF measures of amyloid and tau, along with measures of hippocampal and entorhinal volume, the results show that brain activity is most closely associated with hippocampal volume and tau levels. Brain activity was also associated with a baseline measure of disease stage which represented a compilation of the various markers. The study is well thought out and the analyses are in depth. I particularly appreciate the examination of brain function in relation to the markers of AD, as the symptomology of dementia all comes down to how well the brain continues to work in the face of developing neuropathology. I have the following comments.

R3.01: The disease stage is set on an arbitrary scale of years. Is there a way to anchor the disease stage to the diagnosis year in those with cognitive impairment?

Response to R3.01: We thank the reviewer for their suggestion to anchor the disease stage to the diagnosis year in those with cognitive impairment. We have opted to anchor the disease stage values around the median disease stage of healthy controls who are amyloid positive. This is in line with a recent DELCODE study (Baumeister et al., 2025). Further, this was done, because diagnostic criteria for a clinical diagnosis in DELCODE are based on neuropsychological test evaluations (Jessen et al., 2018). However, it has been shown that biomarker depositions may occur much earlier in the disease cascade than the first cognitive symptoms (Palmqvist et al., 2017, Jack et al., 2016). Thus, after disease stage estimation, we calculated the median disease stages for the biomarker groups. We used the median in the healthy controls with A+T-biomarker status at baseline to anchor the disease stages for the whole sample. In accordance, we made changes to all subfigures containing the disease stage values. Likewise, we have included a paragraph in the results section detailing this procedure.

Results: “The obtained disease stages are probabilistic estimates given the available biomarker and cognitive test data of the patient. Thus, the disease stage values characterize the relative rather than absolute stages reflecting individual differences in the analyzed sample on the latent disease time axis span by the DPM. To aid clinical interpretation, we anchored the disease stages around the median disease stage of healthy controls with amyloid positivity (CN A+T-).” (lines 180 – 185)

R3.02: Figure 1 and other places. The term ‘model posterior’ should be more well defined.

Response to R3.02: We agree with the reviewer that the term was not clear enough. Therefore, we have provided more detail in the beginning of the results section 2.1, with a focus on the Bayesian properties of the DPM .

Results: “In order to obtain a continuous AD-related disease stage score, we used 739 longitudinal ATN biomarker and cognitive measurements from 208 participants to train the probabilistic DP model (see methods for details; Supplementary Table 2 for baseline characteristics of this subsample) including hyperparameter optimization. The results of this were two-fold. First, a posterior probability distribution of the biomarker disease progression curves conditioned on the training data was obtained. For visualization, we plotted the average curves after randomly sampling from this posterior probability distribution 200 times (Fig. 1A, Supplementary Figure 1). Second, each participant was assigned a continuous disease stage value on a new disease progression time axis (with arbitrary offset) in years. The obtained disease stages

are probabilistic estimates given the available biomarker and cognitive test data of the patient and the training data. Importantly, the disease stage values characterize the relative rather than absolute stages reflecting individual differences in the analyzed sample on the latent disease time axis span by the DPM. Concretely, a participant's disease stage is given as the minimum in the negative log-likelihood function over the disease stages given all available participant data. In this regard, more positive disease stages are related to objective memory impairment, amyloid positivity, and conversion from CN/SCD to MCI or AD (Supplementary Figure 2)." (lines 169 – 189)

In order to facilitate understanding even more, we have additionally added the following section in the online methods:

Online Methods: "In detail, according to Lorenzi et al.¹⁴, the estimated disease stage is given as the expected value of

$$p(t^* | y^*, y, t, m, t') = \frac{(p(y^* | t^*, y, t, m, t') * p(t^*))}{(p(y^* | y, t, m, t'))} \text{ (lines 659 – 661).}$$

R3.03: 2.5 Longitudinal activation change is moderated by disease stage. It should be clear that the association between longitudinal change in activation was observed with baseline disease stage. It is also interesting that the baseline analyses shows a relationship, but no relationship is seen with longitudinal disease state change. Why would that be?

Response to R3.03: We thank the reviewer for the careful observation of the data. In response to the first part of the reviewer's comment, we added the following statement to the longitudinal results part. We thereby hope to clarify that the association is indeed, as the reviewer correctly pointed out, based on cross-sectional disease stages.

Results: "To this end, we hypothesized that cross-sectional disease stage would moderate longitudinal changes in successful encoding-related activity across follow-ups, particularly in deactivation." (lines 333-334)

With regards to the non-significant longitudinal change-change association, we would like to point out that the change-change association can be obscured by within-person variability which is often observed in longitudinal fMRI (see Elliot et al., 2022; Mooraj et al., 2024). It has recently been shown that fMRI possesses low to medium test-retest reliability (Elliot et al., 2022). It is therefore not unlikely that the non-significant change-change associations were obscured by constraints from the task-fMRI paradigm itself. It is for this reason that we have added the following section in the results at the position of the random-intercept-random-slope models:

Results: "A potential reason may be the inherent problem of large intra-individual variability known to occur in longitudinal task-fMRI^{45,46}" (lines 340 – 341)

R3.04: Information is needed on the longitudinal data. For example, what is the mean interval of the various longitudinal assessments (CSF, fMR, etc)? This should be added to the text in appropriate places.

Response to R3.04: We thank the reviewer for the helpful suggestion and apologize for the apparent lack of details about the design and nature of the longitudinal data used in this study. Accordingly, we now report mean longitudinal follow-up time in the revised supplement and main body of the manuscript. We have additionally incorporated mean and standard deviation of group-wise longitudinal follow-up time in the supplementary tables 1 & 2. For the analysis sample, we only report on longitudinal fMRI follow-up time, because we did not use longitudinal biomarker data for our analysis other than the estimation of disease stages. Within the results, we added the following paragraphs relating to the data availability:

Results: “In order to obtain a continuous AD-related disease stage score, we used 739 longitudinal ATN biomarker and cognitive measurements from 208 participants with an AD biomarker profile¹¹ to train the probabilistic DP model (see methods for details; Supplementary Table 2 for baseline characteristics and longitudinal follow-up availability of biomarkers of this subsample). Of note, follow-up years for CSF biomarkers, volumetrics, and cognition were longest in CN, followed by SCD, MCI, and DAT, respectively. For statistics of those differences, please see supplementary table 2.” (lines 169-175)

Supplementary Table 1:

	Missing	CN (n = 165)	SCD (n = 214)	MCI (n = 82)	DAT (n = 32)	Group test P-value	Pairwise comparisons
Age (y)	-	69.17 (5.24)	70.51 (6.05)	72.72 (5.65)	73.86 (6.26)	< 0.001	CN vs. SCD, MCI, DAT SCD vs. MCI, DAT
Sex (% female)	-	105 (63.64)	104 (48.60)	43 (52.44)	18 (56.25)	0.041	CN vs. SCD CN vs. DAT
Education (y)	-	14.61 (2.68)	15.28 (2.93)	13.90 (3.02)	13.09 (2.86)	< 0.001	SCD vs. MCI, DAT CN vs. MCI, DAT
APOE ε4 carrier (%)	5 (1.01)	38 (23.60)	70 (32.86)	41 (50.00)	21 (65.63)	< 0.001	SCD vs. MCI, DAT CN vs. MCI, DAT
CSF-Aβ42/40 ratio	275 (55.78)	0.10 (0.02)	0.09 (0.03)	0.07 (0.03)	0.05 (0.02)	< 0.001 [†]	SCD vs. MCI, DAT MCI vs. DAT
CSF-pTau ₁₈₁ (pg/ml)	275 (55.78)	45.86 (13.51)	51.68 (20.50)	68.02 (34.00)	99.06 (42.55)	< 0.001 [†]	CN vs. MCI, DAT SCD vs. MCI, DAT MCI vs. DAT
Follow-up time (years)							
fMRI data	0	1.95 (1.30)	1.73 (1.16)	1.17 (1.04)	.56 (.84)	< 0.001 [†]	CN vs. SCD, MCI, DAT SCD vs. MCI, DAT MCI vs. DAT

The groups were significantly different in terms of age ($F_{3,489} = 10.59, p < .001$), sex ($\chi^2_3 = 8.29, p = .041$), years of education ($F_{3,489} = 8.45; p < .001$) and APOE ε4 carriership ($\chi^2_3 = 31.26, p < .001$). Additionally, CSF-Aβ42/40 ratios ($F_{3,76,297} = 52.17, p < .001$) and CSF-pTau₁₈₁ levels ($F_{3,62,296} = 14.60, p < .001$) were different across the groups. Finally, follow-up time was significantly different across the groups ($F_{3,132,41} = 25.189, p < 0.001$). CN = healthy controls, SCD = subjective cognitive decline, MCI = mild cognitive impairment, DAT = mild dementia of the Alzheimer’s disease type. [†]F-test calculated using the Welch method. Pairwise comparisons denote significant pairwise post-hoc comparisons after correction using Bonferroni-Holm.

Supplementary Table 2:

	Missing	CN (n = 80)	SCD (n = 57)	MCI (n = 44)	DAT (n = 27)	Group test P-value	Pairwise comparisons
Age (y)	-	69.22 (4.84)	72.66 (5.16)	73.84 (5.38)	75.11 (5.57)	< 0.001	CN vs. SCD, MCI, DAT
Sex (% female)	-	41 (51.25)	21 (38.84)	21 (47.73)	16 (59.26)	0.208	-
Education (y)	-	14.36 (2.67)	15.25 (3.02)	13.93 (2.90)	12.74 (2.57)	< 0.001	SCD vs. AD
APOE ε4 carrier (%)	1 (0.5)	19 (23.75)	38 (67.86)	30 (68.18)	17 (62.06)	< 0.001	CN vs. SCD, MCI, DAT
CSF-Aβ42/40 ratio	-	0.10 (0.02)	0.06 (0.01)	0.05 (0.01)	0.05 (0.01)	< 0.001 [†]	CN vs. SCD, MCI, DAT SCD vs. MCI, DAT
CSF-pTau ₁₈₁ (pg/ml)	-	46.83 (14.39)	72.79 (30.06)	84.04 (29.18)	89.61 (28.14)	< 0.001 [†]	CN vs. SCD, MCI, DAT SCD vs. MCI, DAT
Follow-up time (years)							
CSF	421 (56)	1.56 (1.68)	1.24 (1.74)	1.00 (1.74)	.32 (.97)	< 0.001 [†]	CN vs. DAT
MTL Volumes	205 (27)	2.25 (1.16)	1.71 (1.28)	1.35 (1.39)	.63 (1.14)	< 0.001 [†]	CN vs. MCI, DAT SCD vs. DAT
PACC5	89 (12)	3.54 (2.13)	2.50 (1.99)	1.44 (1.87)	.44 (.96)	< 0.001 [†]	CN vs. SCD, MCI, DAT SCD vs. MCI, DAT MCI vs. DAT
ADAS-COG-13	33 (4)	3.53 (2.32)	2.57 (2.03)	2.13 (2.32)	1.16 (1.74)	< 0.001	CN vs. SCD, MCI, DAT SCD vs. AD

The groups were significantly different in terms of age ($F_{3,204} = 13.27, p < .001$), years of education ($F_{3,204} = 5.14, p = .002$), APOE ε4 carriership ($\chi^2_3 = 36.82, p < .001$), CSF-Aβ42/40 ratio ($F_{3,96,875} = 100.71, p < .001$), and CSF-pTau₁₈₁ ($F_{3,73,993} = 42.36, p < .001$). Sex distributions were not different across the groups ($\chi^2_3 = 4.54, p = .208$). For the longitudinal data, CSF ($F_{3,97,152} = 7.6592, p < 0.001$), volume ($F_{3,87,992} = 13.773, p < 0.001$), PACC5 ($F_{3,102,78} = 38.858, p < 0.001$), and ADAS-COG-13 ($F_{3,204} = 10.25, p < 0.001$) follow-up times were significantly different across the groups. Pairwise comparisons denote significant post-hoc comparisons after Bonferroni-Holm correction. CN = healthy controls, SCD = subjective cognitive decline, MCI = mild cognitive impairment, DAT = mild dementia of the Alzheimer’s disease type. [†]F-test calculated using the Welch method.

Additionally, we included the following paragraph in the results section concerning the longitudinal data availability:

Results: *“If longitudinal data were reflective of our empirical disease curves for fMRI from above, this would provide further leverage for establishing fMRI as a biomarker in AD. To this end, we hypothesized that the disease stage variable would moderate longitudinal changes in successful encoding-related activity across follow-ups, particularly in deactivation. For this analysis, we used longitudinal available fMRI data from four time points. Mean follow-up availability was different across the groups with more data available for CN than any other group ($F_{3,132.41} = 25.189, p < 0.001, \eta^2 = 0.10$ [0.06; 1]; Supplementary table 1).”*(lines 333-339)

R3.05: 4.6 fMRI preprocessing and first-level modelling. More info is needed to explain the exact contrast assessed in the analyses. For example, here it says 'the successful memory encoding contrast' was used. What was the baseline for this contrast? The discussion states the differences between remembered and forgotten stimuli, but that is unclear here.

Response to 3.05: We thank the reviewer for highlighting missing relevant information about the fMRI contrast. In reference to our response R2.02, the successful memory encoding contrast can be constructed in two alternative ways. One is the classical categorical way in which the successfully remembered items are contrasted by the later forgotten stimuli. In our case, the successful memory encoding contrast used in this study is a novelty contrast parametrically modulated by the participants' confidence ratings of having seen a novel item during the fMRI experiment. For this contrast, all novel stimuli are aggregated into a single onset regressor. This single onset regressor is then modulated parametrically via the confidence ratings from the recognition task post-fMRI. Therefore, the baseline is an implicit one in which a beta of 0 would indicate no relationship between successful memory encoding and activation.

In response to the request, we have included additional details about the chosen contrast in the online methods:

Online Methods: *“The contrast of interest was the successful memory encoding contrast. It is based on the single onset regressor containing all novel scenes parametrically modulated by the confidence ratings x from the recognition task post-fMRI $PM = \arcsine\left(\frac{x-3}{2}\right) \times \frac{2}{\pi}$. Thus, the positive t -contrast we used for this study shows regions which are more strongly activated (positive numbers) and deactivated (negative numbers) for successfully encoded items.”* (lines 622 – 627)

R3.06: Discussion, paragraph 2. It would be helpful to reiterate what the time points of neuropathologic change are.

Response to R3.06: We thank the reviewer for this helpful suggestion. In accordance, we have included the time points of neuropathological change in the revised discussion section.

Discussion: *“As predicted, disease stage scores were associated with clinical groups, ATN staging and memory accuracy in the fMRI task (A prime) (Fig 1C,D,E). Finally, the inferred time points of fastest change in the data-driven model ($A\beta_{42/40}$: 1.17; $p\tau_{\text{A}\beta_1}$: 2.79; Entorhinal Volume: 4.09; Hippocampal Volume: 5.38; Episodic Memory Deactivation: 6.36; PACC5: 8.95; ADAS-COG-13: 10.57; Fig. 1B) do align with previously hypothesized trajectory patterns⁴¹ as well as findings from a previous DPM study⁴⁸.”* (lines 376 – 381)

Reviewer #4:

Lattmann et al. present a novel application of disease progression modeling to estimate Alzheimer's disease progression stages. The estimated disease stages were then related to longitudinal changes to episodic memory ability as well as localized changes in functional activation/deactivation during memory encoding. This capacity of the methods presented in this work to facilitate new insights into the progression of Alzheimer's disease is evident, however, the manuscript also demonstrated some weaknesses that prevent me from recommending it for publication in its current state.

My primary concerns deal with the presentation of the author's methods, and with some theoretical weaknesses presented in the introduction and/or discussion sections of the paper. Within that first domain, I find the author's presentation of their DPM method to be somewhat opaque to readers not already familiar with that methodology and request that the DPM methodology is more clearly conveyed throughout the manuscript (see comments 3, 16, and 21 below). Also pertaining to methodology, the author's classification of "disease stage" – which many of their analyses and the interpretation of their results rely on – is not made entirely clear in this manuscript (see comments 8-10 and 15). Finally, I feel that the authors' conceptualization of "synaptic dysfunction" makes some unwarranted assumptions about the mechanisms underlying Alzheimer's disease pathology, which should be qualified or corrected (see comments 6, 7, 17-19).

Response: We thank the reviewer for the positive assessments of the manuscripts potential and the helpful suggestions. Among the specific revisions outlined below we have generally revised and extended the methodological presentation in the introduction, results, and online methods to address these requests and improve clarity and transparency.

Specific Comments:

R4.01: Abstract.

a. "loss of deactivation" juxtapose with "hyperactivation" (meaning increase in activation) is confusing, if technically correct. Please re-word for clarity.

Response to R4.01: We thank the reviewer the suggestion to improve clarity in the abstract. Accordingly, we have adjusted the abstract such that potential confusion should now be avoided.

Abstract: *"Alzheimer's disease (AD) is a major cause of dementia and cognitive decline. Here, we assessed how episodic memory network dysfunction, a hallmark of AD, is associated to the progression of AD biomarkers, MTL atrophy and cognitive scores using data from the DZNE DELCODE study. This data set is unique by including over 1000 longitudinal functional magnetic resonance imaging scans of episodic memory network function. Specifically, we explored the association of successful memory encoding fMRI contrasts with individual disease progression scores from a multi-domain disease progression model (DPM). Voxel-wise analyses revealed reduced deactivations and activations with advanced disease progression. The trajectory for disease-related changes of deactivation were found to be nonlinear, associated with amyloid- and tau-positivity and visually preceded trajectories of cognitive decline. The relationship between deactivation and cognitive decline was partly independent of neurodegeneration. Our results provide evidence that synaptic dysfunction and neurodegeneration are independent drivers of cognitive decline, providing a rationale for targeting synaptic dysfunction along the AD cascade."* (lines 56 – 68)

R4.02: p3 "western societies".

a. Why specify western societies here? Surely there is enough evidence to conclude that Alzheimer's disease is of global concern.

Response to R4.02: We thank the reviewer for this comment and apologize for the suboptimal choice of words. As a result, we have changed the introductory sentence to our manuscript and also changed the citation accordingly that now references the expected prevalence increase all around the globe.

Introduction: *“Alzheimer’s disease (AD) is one of the leading causes of dementia and cognitive decline.”* (line 84)

R4.03: p3 “disease progression model (DPM)”.

a. The methodological advancement of this paper hinges on this statistical approach - more effort should be made to describe this modeling approach - how does DPM differ from other longitudinal modeling approaches that have been applied to Alzheimer's disease progression in the past?

Response to R4.03: We appreciate the reviewer’s suggestion to include more details about the DPM into the manuscript. In reference to our response to R1.04, we have added significant detail in the method section to elaborate on this approach. Likewise, we have extended early descriptions of this approach in the results section:

Results: *“In order to obtain a continuous AD-related disease stage score, we used 739 longitudinal ATN biomarker and cognitive measurements from 208 participants with an AD biomarker profile¹¹ to train the probabilistic DP model (see methods for details; Supplementary Table 2 for baseline characteristics of this subsample). The results of this model were two-fold. First, a posterior probability distribution for the empirical biomarker disease progression curves was obtained. For visualization, we plotted the average GPs after randomly sampling from this posterior probability distribution 200 times (Fig. 1A, Supplementary Figure 1). Second, each participant was assigned a continuous disease stage value on an arbitrary scale in years. The obtained disease stages are probabilistic estimates given the available biomarker and cognitive test data of the patient. Thus, the disease stage values characterize the relative rather than absolute stages reflecting individual differences in the analyzed sample on the latent disease time axis span by the DPM. Concretely, a participant’s disease stage is given as the minimum in the negative log-likelihood function over the disease stages given all available participant data. In this regard, more positive disease stages are related to objective memory impairment, amyloid positivity, and conversion from CN/SCD to MCI or AD (Supplementary Figure 2). ”* (lines 169 – 190)

Response to R4.03 (continued): In response to the reviewer’s question regarding the differences of the DPM utilized in this study and previous advances made in longitudinal disease progression modelling, we would like to provide more detail on a comparison on different methods. One of the earliest continuous DPMs using longitudinal data was conceived by Jedynak et al. (2013). They obtained longitudinal multivariate data from the Alzheimer’s Disease Neuroimaging Initiative (ADNI) data set to calculate an Alzheimer’s disease progression score. With this method, a continuous score was introduced to model the participant’s stage and current progression through the disease. However, this model was conceived focusing on the classic sigmoidal shapes of biomarker curves. A more flexible approach was proposed by Villemagne et al. (2013). With the help of differential equations in a Bayesian setting, they could estimate more flexible shapes than the sigmoidal shapes used in Jedynak et al. (2013). This approach was further utilized in a study of familial AD by Oxtoby et al (2018). Another study used self-modelling regression (Donoghue et al., 2014) to model more complex shapes other than the sigmoidal trajectories proposed by Jack et al. (2010). Time information within this model is coded as time-until-symptom-onset (Donoghue et al., 2014). For this relational staging, participants are shifted on a latent time domain to enable a classification of participant’s disease stage on a latent time frame, comparable to the DPM utilized in this study by Lorenzi et al. (2019). We have integrated more specific differences in the revised discussion section.

Discussion: *“The DPM approach we focused on deviates from parametric disease progression models that impose different and often stronger assumptions on trajectories (e.g., sigmoidal, piecewise linear or dynamical approaches; e.g. ^{12,15,57}). The Gaussian process framework of Lorenzi et al. ¹⁴ provides a non-parametric, continuous representation of disease evolution, allowing relatively flexible capture of non-linear*

abnormality curves without predefining functional forms. This distinguishes it also from popular event-based subtyping models such as SuStaln^{68,69}, which infer discrete subtype-specific stage-wise sequences of biomarker abnormalities. Importantly, while many studies have focused on single-domain characterizations (e.g., cortical atrophy only), the approach used here considers progression and staging as a multi-domain process (atrophy, biomarkers, cognition), enabling a more comprehensive account of AD progression. Future studies might incorporate subtyping and continuous multi-domain approaches for the study of functional alterations in AD.” (lines 511 – 524)

R4.04: p3 “longitudinal CSF biomarkers”

a. Example of such biomarkers would be useful

Response to R4.04: We have included examples of longitudinal CSF biomarkers and references which include studies using those biomarkers in their DPM approaches in this part of the introduction.

***Introduction:** “Investigation of such models requires the conjoint availability of longitudinal CSF biomarkers such as amyloid 42/40 or pTau, volumetry and cognitive markers across the whole AD spectrum ranging from cognitively unimpaired to mild cognitive impairment and mild dementia¹². This is highlighted by recent advances from studies using continuous DPMs. Results showed that time frames spanning the whole disease cascade could be estimated from shorter-scale longitudinal data^{13–16}.” (lines 102 – 107)*

R4.05: p3 “It is one of the first cognitive faculties”

a. Ambiguous "It" here - initially unclear if authors are referring to episodic memory or default mode network function.

Response to R4.05: We thank the reviewer for pointing out this ambiguity. In accordance, we have changed the part in the introduction to replace “it” with episodic memory (EM). Further, we have changed the naming after the initial mention of episodic memory and replaced it with “EM”:

***Introduction:** “Episodic memory (EM), the ability to recall recent personal experiences^{16,17}, is critically dependent on the medial temporal lobe (MTL) memory system and structures of the so-called Default Model Network (DMN)¹⁸. EM is one of the first cognitive faculties to be impaired along the AD cascade². Consequently, a major effort of therapeutic and interventional studies is to slow EM decline or even improve EM function in AD through interventions, including lifestyle¹⁹, pharmacology²⁰ or transcranial brain stimulation²¹. Improving synaptic function through such interventions could potentially also ameliorate EM impairment.” (lines 108 – 114)*

R4.06: p3 “The prevailing cascade models of AD in humans are not compatible with this possibility because they consider memory impairment to be a consequence of neurodegeneration rather than synaptic dysfunction”

a. This is not entirely true - cognitive and neurological scaffolding mechanisms allow for the maintenance or repletion of cognitive ability in light of pathological decline, though any gains are likely to be short-lived in cases where neurodegenerative disease is present. See the Park & Reuter-Lorenz STAC model series of papers.

Response to R4.06: We thank the reviewer for the critical assessment of our manuscript. Indeed, the Scaffolding Theory of Aging and Cognition (Park & Reuter-Lorenz, 2009) provides an important contribution to the explanation of cognitive variability across the aging and dementia spectrum. In the latest revision of this model, Park and Reuter-Lorenz (STAC-R; 2024) add lifestyle factors into this model to explain that neural resources can already be built in early life, thereby even enhancing the scaffolding around the aging brain. They introduce the term of cognitive maintenance to highlight that even in the face of pathology, cognition can be preserved to a certain extent. Additionally, the model encompasses functional aspects including compensation allowing for changes on the functional level in the face of pathology. However, while STAC-R and other models deal with the degree of interindividual differences across aging and, to a certain extent, neurodegenerative diseases, it is not targeted at a group-level trajectory of a disease course. We

agree with the reviewer that functional mechanisms such as compensation are mentioned in the STAC-R model. We have therefore re-written parts of the introduction to incorporate scaffolding and other potential mechanisms explaining the variability of cognitive performance.

Introduction: *"However, while synaptic dysfunction may be one potential candidate as explanatory factor of brain activity abnormalities in the human fMRI, further mechanisms have been proposed. For example, studies have provided evidence that vascular pathology³² or functional connectivity³³ can account for BOLD signal changes. Furthermore, in the cognitive neuroscience literature, dedifferentiation³⁴ and scaffolding³⁵ mechanisms are discussed. There, it is still not clear in how far activation abnormalities present as compensatory^{35,36} or detrimental."* (lines 145 – 150)

R4.07: p4 "We defined synaptic dysfunction as a brain activity abnormality that cannot be explained by neurodegeneration"

a. Why "synaptic" in that case? This definition seems to indicate functional changes that are not paired with pathological structural changes... cardiovascular mechanisms and functional connectivity mechanisms are just as likely as mechanisms operating within the synapse. What is the rationale behind emphasizing synaptic dysfunction rather than these other factors or a combination of these factors?

Response to R4.07: We thank the reviewer for this highly relevant point. In line with other reviewers who similarly suggested the potential relevance of additional mechanisms, we would like to refer to our response to R2.01. In this section, we provide additional analyses investigating vascular brain properties as well as connectivity measures as further explanatory variables for mediating the association between disease-related memory encoding contrast and memory decline. These new conducted analyses revealed that the associations remain significant beyond the influence of vascular, connectivity and also atrophy variability in our sample. Thus, although we cannot exclude potential mediating effects of other unobserved variables, synaptic dysfunction within the episodic memory network would be in line with our findings. We have therefore included a further discussion section to discuss additional explanations.

Discussion: *"However, since synaptic dysfunction may not be the only mechanistic explanation behind our results, we additionally investigated the influence of other potential mechanisms including functional connectivity, WMH, and neural noise. We found that, even when including functional connectivity and WMH volume into a parallel mediation model in conjunction with measures of MTL volume, the association between encoding-related activation and cognition remained. Therefore, it appears unlikely that our disease-related activation was driven by white matter damage or functional connectivity. With regards to the investigation of noise or unexplained variability in the fMRI, we found that overlapping disease-related differences in (de-)activation and noise both explained cognitive variability, yet, the effect sizes for BOLD activity were much larger than for noise when incorporated into the same model. Specifically, we found that unexplained variability increased over AD disease stages. Those findings may be related to studies showing that AD is characterized by a reduction in neural selectivity⁵⁴, with potential mechanistic explanations being related to excitation-inhibition imbalance⁵⁵, dedifferentiation³⁴, or a breakdown of scaffolding³⁵ mechanisms."* (lines 429 – 442)

R4.08: p5 "Second, each participant was assigned a continuous disease stage value... Bivariate correlations between DPM markers and obtained disease stages are provided in Supplementary Figure 2."

a. A brief summary of these disease stages would be appropriate here, especially considering the Methods section of this paper is online-only. Was this classification based on Braak's staging for disease progression? Something else?

Response to R4.08: We apologize for the potential lack of clarity in the description in the manuscript regarding the DPM conception. One assumption of our DPM approach is to quantify disease stage in a continuous rather than discrete way (as the name stage might suggest). This is due to the fact that, from the biological perspective on disease progression for many neurodegenerative diseases assuming a continuous process is often useful since there is little empirical evidence for actual separable stages in terms of

quantitative parameters, e.g. as reflected by stage-specific stability of biomarkers for some period and fast transitions across stages (gradients). This might be different to the clinical and purely behavioral perspective on disease-related outcomes but appears to be a generally reasonable and accepted view in context of DPMs (Young et al., 2024). We thus opted for a DPM model which incorporates multiple biomarker domains and assumes monotonic continuous disease progression towards abnormality (as initially suggested in Jack et. al 2010, 2014). In that sense, our approach to quantify disease stage is rather resulting in a disease progression score (or index) with all possible intermediate values which can be well summarized using a histograms or ridge line plots as shown in figure 1 of the manuscript. The interpretation is related to the actual relative position of each individual. In more detail, it is the expected position on the latent disease time frame given the data of the participant. In light of this description, we apologize to the reviewer that a definition of disease stage values did not initially provide sufficient clarity. In fact, the relative position of each subject on the latent disease time frame (i.e. disease stage) was not based on Braak's staging, but on a posterior probability distribution based on the DPM and the relevant biomarker data (CSF, volumetry, and cognition) from the participant. We have included an improved description of the data-driven proxy of a disease stage values in the respective results section to provide more clarity regarding the approach.

Results: “[...] Second, each participant was assigned a continuous disease stage value on an arbitrary scale in years. The obtained disease stages are probabilistic estimates given (A) the available medial temporal lobe volumes, CSF biomarkers as well as cognitive test scores of the test patient; and (B) an empirical progression model of these outcomes in the training sample showing progression from healthy to advanced AD (see methods). Thus, the disease stage values characterize the relative rather than absolute stages reflecting individual differences in the analyzed sample on the latent disease time axis span by the DPM. In detail, more positive disease stages are related to objective memory impairment, amyloid positivity, and conversion from CN/SCD to MCI or AD (Supplementary Figure 2). Bivariate correlations between DPM markers and obtained disease stages are provided in Supplementary Figure 3.” (lines 179 – 190)

R4.09: p5 “disease stage value on an arbitrary scale in years” (3)
a. What does this mean? What does this arbitrary scale correspond to, and why was it equated to years?

Response to R4.09: We regret the missing information about the scale of the DPM. The arbitrary scale comes from the fact that it is not a measured time frame, but rather an estimated time frame given the potential biomarker progression in the fitting sample. It is thus a hidden variable that is inferred based on data. As a potential analogy, we would like to refer to the differential psychology literature with the conceptualization of intelligence scores such as IQ. There, definitions were based on a mean of 100 with a standard deviation of 15. Here, for the DPM we used from Lorenzi and colleagues (2019), the scale has even more degrees of freedom by allowing for a complete free scale estimation based on the fitting of Gaussian Processes to the data. There, the transformation from recorded time (in our case in years) is a linear transformation. This is sensitive to the input (a time frame provided in months would ultimately change the scale and its numbers). The important detail to consider is that the actual values do not bear meaning as such. It is rather the relative position between different participants that provides exceptional utility over other DPMs not anchored in continuous approaches. In reference to a response to the comment 1 by reviewer 3, we have anchored disease stages to the median disease stage value of amyloid positive healthy controls. Thereby, the disease stage value 0 can now be interpreted as the position on the latent disease time frame around which participants are collected which are rather early in the disease cascade in comparison to those who have larger disease stage values, which would then mean that they are further along the disease cascade as this reference group.

Results: “The obtained disease stages are probabilistic estimates given the available biomarker and cognitive test data of the patient. Thus, the disease stage values characterize the relative rather than absolute stages reflecting individual differences in the analyzed sample on the latent disease time axis span by the DPM. To aid clinical interpretation, we anchored the disease stages around the median disease stage of healthy controls with amyloid positivity (CN A+T-).“ (lines 180 – 186)

R4.10: p5 “Specifically, higher age was indicative of a more advanced disease stage... .All further analyses with disease stage consider the age-, sex-, and education-corrected disease stage values.”

a. Other demographic factors, particularly racial origin and socio-economic status, would be valuable to consider if those data are available.

Response to R4.10: We appreciate the reviewer’s suggestion regarding the potential influence of covariates such as racial origin and socioeconomic status (SES) to the presented findings. Unfortunately, racial origin data was not acquired and therefore unavailable in the DELCODE cohort. While direct measures of SES (e.g., income, occupation) were also not available for our study, we included years of education as a covariate in all analyses. Educational attainment is widely recognized as one valid proxy for SES in Alzheimer’s disease research, given its strong associations with income, occupational status, healthcare access, and cognitive reserve. Although education may not capture all dimensions of SES, it serves as an indicator of socioeconomic positioning and thus, in our opinion, mitigates potential confounding by SES-related factors in the current study.

R4.11: p5 “As expected, participants further in disease progression showed worse A’ scores of fMRI task performance... accounting for age, sex, and education.”

a. What tasks were these, and more importantly what cognitive abilities did they correspond to?

Response to R4.11: We thank the reviewer for this question and regret the lack of clarification in the first version of the manuscript. We would like to point to our response to a comment R1.05 made by reviewer 1 to answer this point.

R4.12: p7 “we did not find indications for inverted U-shaped associations between disease stage and successful memory encoding.”

a. Did you test for any other non-linearities aside from an inverted U function?

Supplementary Figure 4

Response to R4.12: We thank the reviewer for this interesting suggestion. Yes, in addition to inverted U-shape we additionally investigated U-shaped associations presented in the figure above (Section B). Here, only regions of deactivation show a U-shaped association with disease stage mostly found in temporo-parietal (right-lateralized), insular and frontal cortices (bilateral) above the threshold of $pFWE < 0.05$. All other regions were non-significant. Thus, we restricted our tests for non-linearities to linear and/or quadratic (polynomial degree 2) associations between fMRI memory encoding contrasts and disease progression stage. We report on those associations in the results and supplementary figure 4

Results: Additional u-shaped associations are reported in supplementary figure 4. (lines 235 – 236)

R4.13: p7 “These later changes were accompanied by changes in hippocampal and entorhinal cortex volume.”

a. How was this coupling statistically verified?

Response to R4.13: We thank the reviewer for the questions and apologize for the lack of clarity in this statement. This coupling was not statistically verified. Rather, we aimed at expressing the overlap with the time point of fastest change in volume. Therefore, we have changed this statement to make it more clear:

Results: “These later changes overlapped visually with changes in hippocampal and entorhinal cortex volume.” (lines 248-289)

R4.14: p9 “Random-intercept random-slope models did not converge, which is why we report on the random intercept models only.”

a. How do you interpret this lack of convergence? Sample vs noise issue? Something else?

Response to R4.14: We thank the reviewer for this important question. During the revision process, we found a typo in the model specification code for the random-intercept-random-slope longitudinal LME models. After error correction, the models converged, but observed that the estimated variance component of the random slope parameters were close to zero (indicating singularity). In our context, this may be related to the intra-individual variability that is inherently large in task-fMRI (see Mooraj et al., 2025). We do interpret this in context of high intra-individual variability, which is also the reason why we only can report on between-subjects effects in our study for the longitudinal part of our analysis. As a result of the revision process, we have altered the results section accordingly, now also incorporating a putative explanation. Additionally, we now report on the random-intercept-random-slope models as well in the supplement.

Results: “Random effect variances in the random-intercept-random-slope models were very close to zero, which is why we report on the random intercept models only. A potential reason may be the inherent problem of large intra-individual variability known to occur in longitudinal task-fMRI^{40,41}. Nevertheless, we report on the model outputs of the random-intercept-random-slope models in the supplement (Supplementary Table 6 and 7, respectively) for completeness. The corresponding tables for the random-intercept models can be found in the supplement as well (Supplementary Tables 8 and 9, respectively).” (lines 339 – 345)

R4.15: Figure 4 caption “The continuous disease stage scores were used to group participants into early and late stages based on the median disease stage (-3.37) in the sample.”

a. What specifically do you define as "early" vs "late" disease stage

Response to R4.15a: We thank the reviewer for bringing up this issue. Upon revision of the manuscript and in connection with another reviewer’s comment (R1.03), we reanalyzed the longitudinal data using the continuous measure of disease stage as in all other analyses. For this, we would like to draw the reviewer’s attention towards our response to R1.03 to comment on this point.

R4.16: p10 “DPM”

a. A re-summation of this methodological approach at the beginning of the discussion section would aid in reader comprehension.

Response to R4.16: We thank the reviewer for the notion that a short summary section about the DPM would aid the reader’s comprehension. Therefore, we have added the following section to the second paragraph in the discussion, in which we discuss the DPM results:

Results: “We started our analysis by first conceptualizing a continuous Bayesian DPM¹³ using longitudinal CSF, volume and cognition data based on the ATN framework. We obtained biomarker progression curves and disease stage scores. The former resulted in a disease time frame which reflects the successive increase in biomarker abnormality of ATN and cognition data in the DELCODE sample. The disease stages provide probabilistic information about the position of a participant on this arbitrary disease time frame given available longitudinal ATN and cognition data. We found that our disease time frame was around 20 years (Fig. 1A,B).” (lines 368 – 374)

R4.17: p11 “Thus, individuals in our sample show less pronounced brain activity differences between successfully memorized and later forgotten stimuli as cognition declines, an observation in line with the recently reported risk-dependent reduction of fMRI subsequent memory effects across the Alzheimer’s risk spectrum”

a. The justification for this conclusion isn't immediately clear from the results presented - a reference to the specific findings which lead to this conclusion would be helpful.

Response to R4.17: We thank the reviewer for pointing out potential inconsistencies in our interpretations of the results presented in our study. We completely agree with the reviewer’s opinion that the current presentation of results does leave room for questions regarding the conclusion drawn in the discussion. Thus, to provide a remedy, we included also a reference to the figure in the discussion.

Discussion: “After establishing an AD progression marker using the DPM framework, we investigated the relationship between fMRI activations during EM encoding and our DPM scores. Globally, our results indicate a conjoint reduction of encoding-related activations and deactivations as individuals progress along the AD trajectory (see Fig. 2). Thus, individuals in our sample show less pronounced brain activity differences between successfully memorized and later forgotten stimuli as cognition declines, an observation in line with the recently reported risk-dependent reduction of fMRI subsequent memory effects across the Alzheimer’s risk spectrum²⁵.” (lines 382 – 389)

R4.18: p11 “episodic memory circuitry dysfunction”

a. What mechanism specifically does this phrase refer to?

Response to R4.18: We regret the lack of clarity in our terminology and thank the reviewer for highlighting this important point. Previous research has shown that episodic memory function relies on a network of brain regions including the hippocampus, parahippocampus, posterior parietal cortex and prefrontal cortex (Nyberg et al., 2010, Allen & Fortin, 2013, Hassabis & Maguire, 2007). Deviations in activity in this network along the disease progression of AD were termed episodic memory circuit dysfunction in the initial version of the manuscript. However, we are also aware of the different levels of granularity in neuroscience, in which circuits are usually referred to when directly investigating neural circuits (Lawn et al., 2023; Roy et al., 2017; Dickerson & Eichenbaum, 2010) rather than large-scale brain networks. This is not possible with fMRI, as the resolution is too coarse to target circuits directly. We thus agree that the term "circuit" may imply a level of anatomical specificity that exceeds the resolution and interpretive scope of task-based fMRI. In our revision, we have replaced “episodic memory circuit dysfunction” with “episodic memory network dysfunction” throughout the manuscript to more accurately reflect the scale of inference supported by our data. Further, all mentions of “circuit” were replaced by “network” throughout the manuscript.

R4.19: p12 “episodic memory circuitry dysfunction”

a. Again, what does this refer to? It is unclear what you mean by “circuit” in this context.

Response to R4.19: We thank the reviewer for this helpful comment and regret any unclear information presented in the manuscript. We would like to resort to our response to the reviewer’s previous comment (response to R4.18) in order to answer this question. There, we hope to have provided a clearer explanation regarding our terminology.

R4.20: Figure 5

- a. This figure could be simplified by replacing the two unidirectional arrows connecting nodes N and E with a single bidirectional arrow.
- b. Also, the left-justified placement of E vs N, implying a later place in the causal change, may be unwarranted considering the bidirectional influence of neurodegeneration and episodic memory function, and the direct effect of Tau on both processes.

Response to R4.20: We thank the reviewer for the careful examination of our results and suggestions for improving Figure 5. We are in line with the argument of unwarranted later placement of episodic memory dysfunction in the causal chain, given that we only have correlational evidence for the association between E and N. For transparency, this placement was done due to the results from section 2.3, in which we highlight the relative positioning of ATN biomarkers, episodic memory circuit dysfunction, and cognition. However, we also agree that this evidence we found is not enough. Hence, we have augmented figure 5 and the results section, such that a) the two unidirectional arrows connecting N and E were replaced with a single bidirectional arrow and b) adjusted the positioning of N vs E, such that they now represent the same position in the causal chain of events from A, over T, to N & E, and finally to C.

Revised Figure 5 (now Figure 6):

Figure 6. An extended ATN model including episodic memory circuit dysfunction. Our results confirm previous hypothetical conceptualizations of an AD cascade with cognitive impairment (C) being preceded by neurodegeneration (N), tau pathology (T) and amyloid pathology (A). Here, we additionally incorporate episodic memory dysfunction (E) as being partly driven by T and partly by N. E may furthermore result in more N. More explanations in the text.

- R4.21:** p16-17, section 4.7 “Disease progression model for staging based on ATN pathology”
- a. This section is crucial for understanding your findings. I encourage you to import as much of a summary of the DPM method as can be accommodated in the introduction/results section as is feasible

Response to R4.21: We thank the reviewer for highlighting the important point of describing details on the DPM approach earlier in the manuscript. We agree with the reviewer that an earlier description would aid the understanding and story presented. Therefore, we have extended the description and included the following section at the beginning of the results section:

Results: “Baseline demographics are displayed in Supplementary Table 1. Participant selection for the DPM fitting sample was motivated by the ATN framework. In order to obtain a continuous AD-related disease stage score, we used 739 longitudinal ATN biomarker and cognitive measurements from 208 participants with an AD biomarker profile¹¹ to train the probabilistic DP model (see methods for details; Supplementary Table 2 for baseline characteristics and longitudinal follow-up availability of biomarkers of this subsample). Of note, follow-up years for CSF biomarkers, volumetrics, and cognition were longest in CN, followed by SCD, MCI, and DAT, respectively. For statistics of those differences, please see supplementary table 1. The model results were two-fold. First, a posterior probability distribution for the empirical biomarker disease progression curves was obtained. For visualization, we plotted the average GPs after randomly sampling from this posterior probability distribution 200 times (Fig. 1A, Supplementary Figure 1). Second, each

participant was assigned a continuous disease stage value on an arbitrary scale in years. The obtained disease stages are probabilistic estimates given the available biomarker and cognitive test data of the patient. Thus, the disease stage values characterize the relative rather than absolute stages reflecting individual differences in the analyzed sample on the latent disease time axis span by the DPM. To aid clinical interpretation, we anchored the disease stages around the median disease stage of healthy controls with amyloid positivity (CN A+T-). Concretely, a participant's disease stage is given as the minimum in the negative log-likelihood function over the disease stages given all available participant data. In this regard, more positive disease stages are related to objective memory impairment, amyloid positivity, and conversion from CN/SCD to MCI or AD (Supplementary Figure 2).“ (lines 168 – 190)

R4.22: p17 “This resulted in a subsample of 208 participants”

a. The vast majority of your analysis are restricted to this smaller participant pool, no? If so, presenting your sample as 1,011 individuals is misleading

Response to R4.22: We appreciate the reviewer's concern and the opportunity to clarify this point. Our study initially considered the full cohort of 1,011 participants. For the final task-fMRI analyses—the main focus of the manuscript—we utilized the subset of 493 participants with usable and quality-controlled task-fMRI data. Only for the preliminary model fitting step (e.g., optimization of parameters), a further subset of 208 participants was used based on all available DELCODE participants. To avoid any confusion, we agree it would be helpful to state this more explicitly in the manuscript. We have revised the relevant section of the Methods and Results to more clearly describe the sample sizes used at each stage of the analysis pipeline. We also ensured that all references to sample size in the manuscript are clearly contextualized. We hope this revision addresses the reviewer's concern and generally improves the manuscripts clarity for all readers.

Online Methods: “In total, 1011 individuals were enrolled in the study. They participated in a range of neuropsychological tests, structural MRI and fMRI (resting-state and/or task-based, diffusion MRI) sessions, as well as cerebrospinal fluid (CSF) biomarker measurements.” (lines 543 – 545)

Among the 1011 participants, we excluded 12 participants who converted to non-amnestic MCI by April 2021, and 23 additional subjects who did not belong to any of the AD-related biomarker groups (see section 4.3). Finally, the group of ADrel was excluded from our analysis, since we did not have any a priori hypotheses regarding this subsample (for fMRI activation patterns in this group, see^{26,27}). The final analysis sample (n = 493) consisted of 165 cognitively healthy controls (CN), 214 participants with SCD, 82 participants with MCI and 32 participants with DAT, all of whom had available baseline fMRI data. (lines 557 – 563)

“For DPM fit, we only included DELCODE participants who had at least one complete measurement occasion of biomarkers¹⁴. Out of those, we excluded participants with a biomarker profile outside the Alzheimer's continuum (e.g., A-T+; see⁷⁶). This was further refined by restricting the clinical AD-risk groups (i.e., SCD, MCI, and DAT) to those with amyloid positivity (A+). Additionally, we used clinical conversion data to exclude participants who later converted to non-amnestic MCI or to non-Alzheimer's type dementia such as dementia with Lewy bodies, Parkinson's disease dementia, semantic dementia, or vascular dementia. This resulted in a subsample of 208 participants (80 CN, 57 A+ SCD, 44 amnestic A+ MCI, 27 A+ DAT).” (lines 664 – 672)